


# Classical nucleation theory of immersion freezing: Sensitivity of contact angle schemes to thermodynamic and kinetic parameters

Luisa Ickes[1], André Welti[1*], and Ulrike Lohmann[1]

[1]Institute of Atmospheric and Climate Science, ETH Zurich, Zurich, Switzerland
[*]now at: Leibniz Institute for Tropospheric Research, Leipzig, Germany

*Correspondence to:* Luisa Ickes
(luisa.ickes@env.ethz.ch)

**Abstract.** Heterogeneous ice formation by immersion freezing in mixed-phase clouds can be parameterized in general circulation models (GCMs) by Classical Nucleation Theory (CNT). CNT parameterization schemes describe freezing as a stochastic process including the properties of insoluble aerosol particles, so called ice nuclei, in the droplets. There are different ways how to describe the properties of aerosol particles (i.e. contact angle schemes), which are compiled and tested in this paper.

The goal of this study is to find a parameterization scheme for GCMs to describe immersion freezing with the ability to shift and adjust the slope of the freezing curve compared to homogeneous freezing to match experimental data.

The results of using CNT are very sensitive to unconstrained kinetic and thermodynamic parameters in the case of homogeneous freezing leading to uncertainties in calculated nucleation rates $J_{\mathrm{hom}}$ of several orders of magnitude. Here we investigate how sensitive the outcome of a parameter estimation for contact angle schemes from experimental data is to kinetic and ther-

modynamic parameters. We show that additional free parameter can mask the uncertainty of $J_{\mathrm{imm}}$ due to thermodynamic and kinetic parameters.

Different CNT formulations are fitted to an extensive immersion freezing dataset as a function of particle diameter ($d$), temperature $T$ and time $t$ for different mineral dust types, namely kaolinite, illite, montmorillonite, microcline (K-feldspar) and Arizona test dust. It is investigated how accurate different CNT formulations (with the estimated fit parameters) reproduce the

measured freezing curves, especially the time and particle size dependence of the freezing process. The results are compared to a simplified deterministic freezing scheme. It is evaluated in this context which CNT based parameterization scheme to represent particle properties is a good choice to describe immersion freezing in a GCM.

## 1  Introduction

In mixed-phase clouds freezing of cloud droplets occurs by different pathways of heterogeneous freezing/nucleation. The

nucleation process is initiated on the surface of an aerosol particle, called ice nucleus (IN), which either collides with a supercooled droplet (contact freezing), acts as cloud condensation nucleus (CCN) and causes freezing when the droplet is increasingly supercooled (immersion freezing), freezes immediately after CCN activation at supercooled conditions (condensation freezing), or provides a site where water vapor deposits as ice (deposition nucleation) (Vali, 1985).

In mid latitudes, where supercooled clouds are common, IN and their effect on precipitation formation through immersion





freezing influence the hydrological cycle (Lohmann, 2002; Zeng et al., 2009; DeMott et al., 2010) and thereby e.g. the biosphere and agriculture. Aerosol particles determine the formation and ice-water ratio of mixed-phase clouds, thereby the cloud radiative properties and indirectly the radiation budget, which affects earth's climate. Therefore results of climate simulations in regional and global models are sensitive to the parameterization scheme used for heterogeneous ice formation and in partic-
ular immersion freezing as it is the most abundant freezing pathway (Ansmann et al., 2009; Wiacek et al., 2010). One approach to parameterize immersion freezing in global and regional climate models is by Classical Nucleation theory (CNT). CNT is a theory based on approximations considering the thermodynamics and kinetics of nucleation. Although computationally more expensive in most cases compared to empirical parameterization schemes, it allows a physical treatment of ice nucleation as function of temperature $T$, ice supersaturation $S_i$, time $t$ and IN type (e.g. size, surface properties). Using a theoretical scheme
has the advantage that the scheme is valid over the whole $T$-$S_i$-space, which is mandatory for the use in a GCM, where all kind of conditions occur (especially in certain regions, but also in simulations of future climate, where atmospheric conditions can be different from the present day or pre-industrial ones). Empirical schemes are in contrast often limited to narrow conditions the scheme was estimated for and can lead to unphysical results when extrapolated. Therefore empirical schemes might not hold for future atmospheric or untypical atmospheric conditions. One example is the Meyers et al. (1992) scheme, which was
developed using measurements in mid-latitude and has problems when being extrapolated to Arctic conditions (Prenni et al., 2007).

The framework of CNT is partly unconstrained and therefore very sensitive to the choice of thermodynamic and kinetic parameters, namely interfacial tension between ice and water $\sigma_{iw}$ and activation energy $\Delta g^{\#}$. Sensitivity of CNT on $\sigma_{iw}$ and $\Delta g^{\#}$ in the case of homogeneous freezing has been discussed in Ickes et al. (2015). Using CNT as an approach to parameterize
immersion freezing in aerosol-climate models raises the question of the sensitivity of the parameterization scheme to $\sigma_{iw}$ and $\Delta g^{\#}$ in the case of heterogeneous freezing. Additionally there is a need to include and represent IN properties. Here we use three different schemes to describe the effect of an IN population on immersion freezing and investigate the impact of the chosen scheme on the parameterization of immersion freezing. We also discuss strategies how to judge/evaluate different CNT formulations.

The formalism of CNT for immersion freezing is explained in section 2. Advantages and disadvantages of certain formulation for the use in GCMs are discussed. In section 3 the sensitivity of the immersion freezing nucleation rate $J_{imm}$ [s$^{-1}$m$^{-2}$] and the fit of the geometric term $f$ to thermodynamic and kinetic parameters is investigated by fitting and comparing the results to an ice nucleation measurement dataset of kaolinite (Welti et al., 2012). The section is followed by suggestions for criteria how to evaluate the quality of a CNT parameterization scheme (section 4). Finally in section 5 CNT parameters are estimated
from experimental data for five different mineral dust types and afterwards in section 6 the criteria are tested for three CNT parameterization schemes and compared to an empirical parameterization.





## 2 Classical Nucleation Theory for immersion freezing

The presence of IN immersed in supercooled droplets facilitates ice nucleation compared to homogeneous nucleation by providing a catalytic surface. The IN surfaces reduces the thermodynamic energy barrier $\Delta G$ determined by $T$, $S_i$ and $\sigma_{iw}$. The difference in nucleation with and without an IN i.e. homogeneous or heterogeneous nucleation, is accounted for by the geometric term $f$, also called wettening factor, compability factor or contact parameter. This term indicates the increased probability to nucleate a stable ice germ due to the presence of the IN surface and therefore reduced number of water molecules necessary to form an ice germ. It describes by how much the IN properties (of unknown nature) reduce the energy barrier for the formation of ice embryos on its surface compared to homogeneous freezing and can be expressed as a function of the contact angle $\alpha$, which is the tangential angle between the ice embryo on the IN surface and the parent phase (here supercooled water) [Fletcher (1958)]:

$$
\begin{aligned}
\Delta G &= f(\alpha) \cdot \Delta G_{\text{hom}} \\
&= f(\alpha) \cdot \frac{16\pi}{3} \cdot \frac{v_{\text{ice}}^2 \sigma_{\text{iw}}^3}{(k_{\text{B}} T \ln S_i)^2}
\end{aligned}
\tag{1}
$$

with

$$
\begin{aligned}
f(\alpha) = \ \frac{1}{2}\Bigg[ &1 + \left(\frac{1 - X \cdot \cos\alpha}{g}\right)^3 + X^3 \left(2 - 3\left(\frac{X - \cos\alpha}{g}\right)\right. \\
&\left. + \left(\frac{X - \cos\alpha}{g}\right)^3\right) + 3 \cdot \cos\alpha \cdot X^2 \left(\frac{X - \cos\alpha}{g} - 1\right) \Bigg]
\end{aligned}
\tag{2}
$$

with

$$
X = \frac{r_{\text{IN}}}{r_{\text{germ}}} \quad \text{and} \quad g = \sqrt{1 + X^2 - 2 \cdot X \cdot \cos\alpha} \, ,
$$

where $v_{\text{ice}}$ is the volume of a water molecule in the ice embryo, $k_{\text{B}}$ is the Boltzmann constant, $S_i$ is the saturation ratio with respect to ice, $r_{\text{IN}}$ is the radius of the catalytic IN surface and $r_{\text{germ}}$ is the critical radius of an ice cluster which initiates freezing of the droplet.

The contact angle $\alpha$ has a value between $0°$ and $180°$, where the latter is equal to the case of homogeneous freezing ($f$=1). If the radius of the IN is significantly larger than the ice germ, radius curvature of the IN surface can be neglected leading to a simplified form of $f$ (Volmer, 1939):

$$
f(\alpha) = \frac{(2 + \cos\alpha)(1 - \cos\alpha)^2}{4} \, .
\tag{3}
$$

Whereas the thermodynamic term in the nucleation rate $J_{\text{imm}}$ (thermodynamic exponent determined by the energy barrier $\Delta G$, see above) changes from homogeneous to heterogeneous freezing, the kinetic term is assumed to be the same for homogeneous and immersion freezing. The kinetics give the number of molecules, which can potentially be incorporated into the ice germ. They are captured in the prefactor of the nucleation rate (see Eq. 4) and the kinetic exponent (determined by the activation energy barrier $\Delta g^{\#}$). The prefactor of the nucleation rate is different in the case of immersion freezing compared to





homogeneous freezing:

$$C_{\mathrm{prefac,hom}} = n_{\mathrm{s}} \cdot 4\pi r_{\mathrm{germ}}^2 \cdot Z \cdot k_B T / h \cdot N_1,$$

where $n_{\mathrm{s}}$ is the number of water molecules in contact with the unit area of the ice cluster, $Z$ is the non-equilibrium Zeldovich factor, $h$ the Planck's constant and $N_1$ is the volume-based number density of water molecules in the liquid parent phase. The difference is due to homogeneous freezing being a volume-dependent process while immersion freezing is assumed to be a surface dependent process. When calculating the non-equilibrium Zeldovich factor $Z$, the freezing type has to be considered:

$$Z = \frac{1}{n_{k,\mathrm{germ}}} \cdot \sqrt{\frac{\Delta G}{3\pi k_{\mathrm{B}} T}} \ .$$

$Z$ is not the same for homogeneous and heterogeneous freezing, because the number of the water molecules in the ice germ, $n_{k,\mathrm{germ}}$, differs. As shown in Pruppacher and Klett (2000) most of the prefactors cancel out in the case of heterogeneous freezing leading to the following expression for the nucleation rate for immersion freezing:

$$J_{\mathrm{imm}}[\mathrm{m}^{-2} \cdot \mathrm{s}^{-1}] \quad = \quad n_{\mathrm{s}} \cdot \frac{k_B T}{h} \cdot \exp\left(-\frac{\Delta g^{\#}}{k_{\mathrm{B}} T}\right) \cdot \exp\left(-\frac{f(\alpha) \cdot \Delta G}{k_{\mathrm{B}} T}\right) \ . \tag{4}$$

Since the energy barrier of immersion freezing is reduced compared to homogeneous freezing, the freezing curve is shifted to higher temperatures and is less steep. This curve shift and flattening is described by the geometric term $f$ and has to be captured by the different CNT formulations.

## 2.1 Parameterization schemes for the geometric term $f$

Different schemes have been put forward to describe the influence of an IN on the nucleation process, i.e. to describe the ice nucleating surface properties of aerosol particles (Marcolli et al., 2007; Lüönd et al., 2010). Thus when fitting experimental data the fit parameter(s) describe the physical properties of the IN. Depending on the scheme these properties are represented by one or several fit parameters and the complexity for an implementation in a GCM differs accordingly. Note that increasing complexity normally comes with higher computational costs.

Three schemes including one or two fit parameters are used in the following sensitivity analysis (section 3) and briefly explained here. A graphical representation of each scheme is shown in Fig. 1. For more details see Marcolli et al. (2007) and Lüönd et al. (2010).

From immersion freezing measurements the frozen fraction $FF$ is obtained, which is the fraction of a droplet population/activated aerosol population that is frozen at a certain temperature $T$ after a certain time $t$. To compare different CNT based parameterization schemes to measurements, $FF$ is calculated from the nucleation rate $J_{\mathrm{imm}}$. The frozen fraction $FF$ is given by:

$$FF = 1 - \exp(-J_{\mathrm{imm}}(T,\alpha) \cdot A_{\mathrm{IN}} \cdot \Delta t), \tag{5}$$

with $A_{\mathrm{IN}}$ being the surface area of the IN. For simplicity particles are assumed to be spherical ($A_{\mathrm{IN}} = 4\pi r_{\mathrm{IN}}^2$). Thus, the surface used for the IN of specific mass represents a lower limit (non-spherical surface would be larger).



### 2.1.1 Single-$\alpha$ scheme

The single-$\alpha$ scheme is assigning one contact angle to the entire surface of each particle. It is based on the assumption that all particles have one common occurrent surface property responsible for their ice nucleating ability. Consequently all particles have an equal probability to act as IN at given conditions. The scheme requires only one fit parameter ($f$ or $\alpha$).

It is the least complex and consequently the cheapest scheme suitable to implement in GCMs. However, it does not take into account that ice nucleating properties might be variable throughout a particle population. This scheme is used in several models, e.g. Khvorostyanov and Curry (2000, 2004, 2005); Liu et al. (2007); Eidhammer et al. (2009); Hoose et al. (2010); Storelvmo et al. (2011); Ervens and Feingold (2012).

### 2.1.2 $\alpha$-pdf scheme

The $\alpha$-pdf scheme is an extension of the single-$\alpha$ scheme. It accounts for the heterogeneity of particles in an aerosol population by using a log-normal probability density function (pdf) for the contact angle $\alpha$. The log-normal distribution of $\alpha$ within a particle population is expressed by two fit parameters, the mean contact angle $\mu$ and the variance $\sigma$ of the distribution:

$$p(\alpha) = \frac{1}{\alpha\sqrt{2\pi\sigma^2}} \cdot \exp\left(-\frac{(\ln(\alpha)-\mu)^2}{2\sigma^2}\right) . \tag{6}$$

This approach attributes an individual surface property to each particle on the entire particle surface.

The variance $\sigma$ defines the heterogeneity of the particle property within the population: the larger the variance $\sigma$, the larger the heterogeneity among the particles. The approach has been used to interpret freezing data, e.g. Marcolli et al. (2007); Lüönd et al. (2010); Broadley et al. (2012); Ervens and Feingold (2012); Welti et al. (2012); Wheeler et al. (2014), because it better represents the nature of the IN-sample. Due to the increased complexity compared to the single-$\alpha$ scheme, only a few attempts have been made to implement it in GCMs [e.g. Wang et al. (2014)]. Application of the scheme in GCMs faces the problem of

the unknown time evolution of the contact angle distribution. Because the most efficient IN will form ice first the remaining contact angle distribution (IN, which did not freeze yet) changes in case an aerosol population is not replenished within one timestep. Without an explicit treatment of the time evolution of the $\alpha$-pdf, ice formation will be overestimated since the most efficient IN can initiate freezing over and over again. Note that this issue is closely connected to the time resolution of the GCM, which will be discussed in a future publication.

The frozen fraction $FF$ is derived by integrating the contact angle distribution over all possible contact angles:

$$FF = 1 - \int_0^\pi p(\alpha) \cdot \exp(-J_{\text{imm}}(T,\alpha) \cdot A_{\text{IN}} \cdot \Delta t) \, \mathrm{d}\alpha . \tag{7}$$

Another extension and frequently used scheme is the active sites scheme, e.g. in Marcolli et al. (2007); Lüönd et al. (2010); Niedermeier et al. (2011); Welti et al. (2012); Wheeler et al. (2014). It goes one step further and assumes several surface sites

on a single IN. Freezing is described based on active sites (initiating the nucleation process), which are randomly distributed on each IN surface within the particle population. As this scheme is computationally too expensive for the use in GCMs it is





left out of the following analysis. More information about the active sites scheme can be found in Marcolli et al. (2007); Lüönd et al. (2010); Niedermeier et al. (2011).

### 2.1.3 Temperature dependent single-$\alpha$ scheme ($\alpha(T)$ scheme)

The $\alpha(T)$ scheme is a compromise between the single-$\alpha$ and the $\alpha$-pdf scheme. It does not take into account how contact
angles are distributed among a particle population but it is assumed that $\alpha$ is different for different $T$, which reflects a change of the $\alpha$-pdf distribution and with that a change in $\mu$ with supercooling or time. This refers to the situation where good IN freeze first at highest temperatures shifting the mean contact angle $\mu$ of the remaining IN population to less efficient IN with further cooling (assuming the aerosol population does not substantially change while cooling and the contact angles are not replenished from one to the next timestep). The $\alpha(T)$ scheme is thus representing the shifted mean contact angle of an initial contact angle
distribution. The temperature dependence of $\alpha$ can be approximated to be linear as discussed in Welti et al. (2012). This scheme is computationally cheaper compared to the $\alpha$-pdf scheme, because no integration over a contact angle distribution is necessary. It also circumvents the issue of shifting $\alpha$-pdf with time, as this is inherently captured in the scheme. Being capable to describe a variability of the freezing process due to a contact angle distribution without being computationally complex makes the $\alpha(T)$ scheme attractive for GCMs. However it demands an indirect assumption on how the aerosol population changes with time or
supercooling, respectively.

The frozen fraction $FF$ is estimated analogously to the single-$\alpha$ scheme using a linear function for $\alpha(T)$:

$$FF \quad = \quad 1 - \exp(-J_{\mathrm{imm}}(T, \alpha(T)) \cdot A_{\mathrm{IN}} \cdot \Delta t) \,, \tag{8}$$

with

$$\alpha(T) \quad = \quad \alpha_0 + m \cdot T \,.$$

## 3   Sensitivity analysis

### 3.1   Fitting immersion freezing measurements

In this section, the sensitivity of $J_{\mathrm{imm}}$ and $FF$ to different combinations of $\sigma_{\mathrm{iw}}$ and $\Delta g^{\#}$ (see Ickes et al., 2015 for a discussion of these parameters) in combination with the contact angle schemes discussed in section 2.1 is analyzed by fitting and comparing the different CNT parameterization schemes to experimental data. This helps to understand how fit parameters influence
the calculated $FF$ curves.

The experimental data taken from Welti et al. (2012) consists of optically detected frozen fractions $FF$ of droplets containing single immersed, monodisperse kaolinite (Fluka) particles. The data consists of $FF$ as a function of $T$, the particle radius $r_{\mathrm{IN}}$ and the residence time in the measurement setup $t$. Experiments were performed using a CFDC (ZINC/IMCA) [see Welti et al. (2012) for more details]. The error bars of the data reflect the uncertainty in the distinction of water droplets and ice
crystals in the detection unit. For the sensitivity analysis the dataset measured after 10 s for kaolinite particles with a diameter of 400 nm is used. Note that the size of the particles might be underestimated due to the assumption of sphericity and therefore



the calculated nucleation rates $J_{\mathrm{imm}}$ from experimental frozen fractions are always the lowest estimate.

To explore the sensitivity of $J_{\mathrm{imm}}$ and $FF$ to thermodynamic and kinetic parameters of CNT we use different CNT formulations. The thermodynamic and kinetic parameters of CNT $\sigma_{\mathrm{iw}}$ and $\Delta g^{\#}$ used here emerged from Ickes et al. (2015). In the following all approaches and different CNT formulations, which are used for the analysis, are listed. An overview is given in

Table 2. Capital letters indicate the author from whose publication thermodynamic and kinetic parameters are used.

#### #1: **Single-$\alpha$ R&D + Z scheme**

The first approach is to use a single-$\alpha$ scheme in combination with the thermodynamic and kinetic parameters shown to be in good agreement with homogeneous nucleation rates [see Ickes et al. (2015)]. When using a single-$\alpha$ scheme it is important that the kinetic and thermodynamic parameters are a combination which reproduces the homogeneous data

well as there is only one fit parameter and uncertainties cannot be compensated by additional parameters. The emerged best fitting combination of $\sigma_{\mathrm{iw}}$ and $\Delta g^{\#}$ (derived independently of $J$) from the analysis of homogeneous data is $\sigma_{\mathrm{iw}}$ from Reinhardt and Doye (2013) and $\Delta g^{\#}$ from Zobrist et al. (2007). It is tested if these formulations of $\sigma_{\mathrm{iw}}$ and $\Delta g^{\#}$ are also applicable to reproduce heterogeneous nucleation rates with assumptions of the single-$\alpha$ scheme.

#### #2: **Single-$\alpha$ R&D + fit $\Delta g^{\#}$ scheme**

An second approach which emerged from Chen et al. (2008) is using a constant $\Delta g^{\#}$ as an additional fit parameter instead of taking a temperature dependent formulation. This assumption might be wrong in the context of homogeneous freezing especially at very low $T$ (Barahona, 2015). However, it should be applicable for immersion freezing as the change in $\Delta g^{\#}$ is small in the corresponding temperature range. The approach is used in combination with $\sigma_{\mathrm{iw}}$ from Reinhardt and Doye (2013) and a single-$\alpha$ scheme. To decide if $\sigma_{\mathrm{iw}}$ from Reinhardt and Doye (2013) is the best choice,

different expressions for $\sigma_{\mathrm{iw}}$ (derived independently of $J$) are tested against a fit of $J_{\mathrm{hom}}$ using constant $\Delta g^{\#}$ (see Fig. 2 analog to Fig. 17 in Ickes et al.,2015). We find that $\sigma_{\mathrm{iw}}$ from Reinhardt and Doye (2013) remains an appropriate choice even when $\Delta g^{\#}$ is used as a constant.

#### #3 + #4 **Single-$\alpha$ O + fit $\Delta g^{\#}$ scheme and single-$\alpha$ E + fit $\Delta g^{\#}$ scheme**

For the sensitivity study of $J_{\mathrm{imm}}$ to the kinetic and thermodynamic parameters the focus is on $\sigma_{\mathrm{iw}}$. To capture the whole

possible range, two formulations of $\sigma_{\mathrm{iw}}$ are used. One from Eadie (1971) leading to the lowest homogeneous nucleation rate and a second formulation of $\sigma_{\mathrm{iw}}$ from Ouchi (1954) leading to the highest homogeneous nucleation rate (see Fig. 2). For a summary of the two formulations of $\sigma_{\mathrm{iw}}$ we refer to Ickes et al. (2015).

These two extremes of $\sigma_{\mathrm{iw}}$ are used together with a constant $\Delta g^{\#}$ (fit parameter) and the single-$\alpha$ scheme to clarify if a fit of $\alpha$ can compensate for a low/high $\sigma_{\mathrm{iw}}$.

#### #5 $\alpha$-pdf R&D + Z scheme

This scheme is an $\alpha$-pdf scheme using the same thermodynamic and kinetic parameter as the single-$\alpha$ scheme (#1). This increases the complexity and adds an additional fit parameter compared to the single-$\alpha$ R&D + Z scheme (#1). By doing





this we test the influence of the choice of the contact angle scheme on the fit result. Additionally we examine if the number of free fit parameters plays a role when choosing a contact angle scheme.

#6 $\alpha$-pdf E + Z scheme

One extreme formulation of $\sigma_{\mathrm{iw}}$ from Eadie (1971) is used together with $\Delta g^{\#}$ from Zobrist et al. (2007) and the $\alpha$-pdf scheme (see 5.).

#7 $\alpha(T)$ R&D + Z scheme

Similar to the $\alpha$-pdf R&D + Z scheme (#5), thermodynamic and kinetic parameter from the single-$\alpha$ scheme (#1) are used with an $\alpha(T)$ scheme instead of the single-$\alpha$ scheme.

The dataset from Welti et al. (2012) is fitted with the previous listed CNT formulations. The fits are done by least-square minimization of $FF$ as a function of $T$. The results are shown in Fig. 3 and Table 2. Table 2 additionally contains a summary of the CNT formulations. Overall most CNT formulations are able to capture the freezing curve well with similar root mean square errors (RMSE) of the estimated and measured freezing curve independent of the thermodynamic and kinetic parameters. In the following the results are discussed in more detail.

The single-$\alpha$ R&D + Z scheme (#1) poorly captures the experimental data and results in a too steep freezing curve. With the single-$\alpha$ scheme it is not possible to reproduce the reduction of the energy barrier in a correct manner and to decrease the temperature dependence of the nucleation rate from the homogeneous to the heterogeneous case. Having only one fit parameter, which in this case is a factor in the exponential term, is not sufficient to shift and flatten the freezing curve compared to homogeneous freezing. Only the $T$-shift of the freezing curve compared to homogeneous freezing is captured by the fitted single-$\alpha$ scheme. This can be seen in a more general illustration in App. A. This result indicates that the simplified assumption of a homogeneous (single) contact angle for the entire population is not sufficient.

Using $\Delta g^{\#}$ as an additional fit parameter (scheme #2) reduces the steepness of the curve. In this case both fit parameters are factors in the exponential term of the nucleation rate with a similar influence on the fitted $FF$. The fit parameter which is multiplied with the temperature dependent variable $\Delta G$ ($f$) mainly shifts the freezing curve but cannot reduce the steepness sufficiently at the same time (see single-$\alpha$ scheme). Using a second fit parameter $\Delta g^{\#}$ resolves this issue. A simplified view on this is, that one fit parameter is responsible for the shift, the other one for the flattening of the immersion freezing curve compared to homogeneous freezing. Using a constant $\Delta g^{\#}$ might be reasonable based on the results from the homogeneous freezing analysis (Ickes et al., 2015), but fitting $\Delta g^{\#}$ to immersion freezing data leads to substantially higher $\Delta g^{\#}$ than those estimated by theoretical calculations (see Ickes et al., 2015). Moreover the fit value of $\Delta g^{\#}$ is aerosol-specific. This might be an artificial result and it is questionable if the assumption of a temperature independent and aerosol type specific (due to the fitting) $\Delta g^{\#}$ is a physical valid approach. It contradicts the assumption that the kinetic parameters such as $\Delta g^{\#}$ are the same for homogeneous and heterogeneous nucleation. The general approach to take the same thermodynamic and kinetic parameters (besides $f$ and the prefactor) for homogeneous and heterogeneous nucleation is based on the assumption that mechanisms in the supercooled water are not influenced by the immersed aerosol particle. This hypothesis might not be true. The aerosol





might influence e.g. the diffusion of water molecules close to the particle which could explain a change in $\Delta g^{\#}$ depending on aerosol type.

An alternative to having $\Delta g^{\#}$ as an additional fit parameter is to use a more sophisticated contact angle scheme, e.g. the $\alpha(T)$ scheme (CNT #5 and #6) or the $\alpha$-pdf scheme (CNT #7). Both approaches lead to good agreement with the experimental freezing curve and can be physically justified because they resemble the natural variability of IN by assuming a contact-angle distribution (in the case of $\alpha(T)$ only indirectly). The $\alpha(T)$ scheme has the disadvantage that it is not known how $\alpha$ changes with $T$. A wrong assumption could lead to an unphysical contact angle scheme, where the change of $\alpha$ does not represent the shift in the contact angle distribution correctly.

The curves resulting from scheme #3-7 (single-$\alpha$ + fit $\Delta g^{\#}$ schemes, $\alpha$-pdf schemes and $\alpha(T)$ scheme) all support the hypothesis that increasing the number of fit parameters from one to two allows to find a reasonable fit, independent of the kinetic and thermodynamic parameters chosen and also independently of the contact angle scheme. This hypothesis is also supported by the result of the single-$\alpha$ scheme, where one fit parameter alone cannot shift and flatten the freezing curve. However, using a single-$\alpha$ scheme with a different additional fit parameter (e.g. the slope of $\sigma_{\mathrm{iw}}$ instead of a constant $\Delta g^{\#}$) does not lead to a better fit of the freezing curve. This might be due to the formula for the energy barrier preventing a sufficient influence of the additional fit parameter on the steepness of the curve. $\Delta g^{\#}$ as an additional fit parameter is able to reduce the steepness of the freezing curve as it has the opposite temperature dependence to the energy barrier $\Delta G$. For a visualization of how the fit parameters influence the freezing curve for each scheme see App. A.

Using one CNT formulation, e.g. the single-$\alpha$ R&D + fit $\Delta g^{\#}$ scheme (#2) together with wrong fit parameters emerging from a fit from a different CNT formulation, e.g. the single-$\alpha$ E + fit $\Delta g^{\#}$ scheme (#4) leads to a wrong freezing curve. This is illustrated in Fig. 3 [solid red line, single-$\alpha$ R&D + fit $\Delta g^{\#}$(E) (#2/4)].

Being able to reproduce experimental data does not directly depend on $\sigma_{\mathrm{iw}}$ and $\Delta g^{\#}$ and not only on the number of fit parameters used, but also on the contact angle scheme. In a reverse conclusion this means that the fit results of $\alpha$, $\alpha_0$ and $m$ or $\mu$ and variance $\sigma$ strongly depend on the CNT formulation used. Looking at Table 2 one can see that $f$ differs substantially, e.g. when using the single-$\alpha$ + fit $\Delta g^{\#}$ scheme with different assumptions of $\sigma_{\mathrm{iw}}$. Comparing $\sigma_{\mathrm{iw}}$ from Reinhardt and Doye (2013) with $\sigma_{\mathrm{iw}}$ from Ouchi (1954) leads to a difference in fitted $f$ of more than 300%, which translates into a difference in contact angle $\alpha$ of approx. 75°. However, all single-$\alpha$ + fit $\Delta g^{\#}$ schemes result in a nearly similar freezing curve with the same RMSE. The fit parameters from the contact angle scheme compensate inaccuracies coming from thermodynamic and kinetic parameters and thus mask potentially wrong assumptions, e.g. of the most important, unconstrained parameter in CNT of homogeneous freezing, $\sigma_{\mathrm{iw}}$. This makes it challenging to compare fit parameters from different studies if not the same CNT formulation was used. Hence in the next subsection we investigate how fit results vary when thermodynamic and kinetic parameters differ and if there is a possibility to compare fit parameters from different studies using different CNT formulations.



## 3.2 Uncertainty of fitting $\alpha$

Table 2 shows that, dependent on the choice of $\sigma_{\mathrm{iw}}$ and $\Delta g^{\#}$, the estimated fit parameters differ. The choice of thermodynamic and kinetic parameters ($\sigma_{\mathrm{iw}}$ and $\Delta g^{\#}$) influences the fit results of different contact angle schemes, which makes comparisons of studies difficult. A contact angle estimate can be different when using CNT with e.g. $\sigma_{\mathrm{iw}}$ from Pruppacher and Klett (2000) compared to using CNT with $\sigma_{\mathrm{iw}}$ from Zobrist et al. (2007). In this section the sensitivity of two contact-angle schemes to $\sigma_{\mathrm{iw}}$ and $\Delta g^{\#}$ is investigated.

The two CNT formulations used in this analysis are a single-$\alpha$ R&D + fit $\Delta g^{\#}$ scheme (#2) and the $\alpha$-pdf R&D + Z scheme (#5) described above. We chose these two schemes because scheme #2 is used in GCMs and scheme #5 to interpret data. Both schemes contain two fit parameters ($f$ and $\Delta g^{\#}$ in scheme #2, $\mu$ and variance $\sigma$ of the contact angle distribution in scheme #5).

We analyze how these two fit parameters depend on a change in thermodynamic and kinetic parameters. For this purpose the thermodynamic and kinetic parameters are varied up to $\pm$ 50%. For the scheme #2 the thermodynamic parameter $\sigma_{\mathrm{iw}}$ is varied, for the scheme #5 the thermodynamic and kinetic parameters $\sigma_{\mathrm{iw}}$ and $\Delta g^{\#}$ are varied separately. The resulting fits are then compared to the reference fit results of section 3.1 (Table 2).

For each variation (e.g. an increase of $\sigma_{\mathrm{iw}}$ by 10%) fitting is done to the same immersion freezing data from Welti et al. (2012) as in the previous section. Figure 4a) shows the relative change of the fit parameters as a function of percentual change in $\sigma_{\mathrm{iw}}$ for scheme #2. The higher the variation of $\sigma_{\mathrm{iw}}$ the larger is the deviation in $f$ from the estimated fit parameters to the reference fit value, whereas $\Delta g^{\#}$ remains unchanged. Fig. 4b) shows the relative change of the fit parameters as a function of percentual change in $\sigma_{\mathrm{iw}}$ and $\Delta g^{\#}$ for scheme #5. In both cases a similar change in fit parameters can be seen. Changing the thermodynamic parameter $\sigma_{\mathrm{iw}}$ has a stronger impact on the fit parameters than changes in the kinetic parameter $\Delta g^{\#}$ [see scheme #5 (Fig. 4 b)]. This is expected from the nucleation rate formula, where $\sigma_{\mathrm{iw}}$ enters the calculation of the nucleation rate to the power of three and therefore changes the nucleation rate/frozen fraction more drastically than a change in $\Delta g^{\#}$.

In case $\sigma_{\mathrm{iw}}$ is increased/overestimated, the fit parameters are decreasing to compensate the change (see dashed arrow in Fig. 4a) and conversely (see dotted arrow in Fig. 4a). The behavior of this compensation is not symmetric but follows the structure of the nucleation rate formula, i.e. $1/x$ dependence for $f$ or $\mu$ and variance $\sigma$, respectively. That implies that the change in the fit parameter gets larger the larger the variation of $\sigma_{\mathrm{iw}}$ is. The relative change approaches -100% with increasing $\sigma_{\mathrm{iw}}$. A larger deviation can be seen for the case where $\sigma_{\mathrm{iw}}$ is decreased/underestimated.

Note that in scheme #2 (Fig. 4 a) only one fit parameter ($f$) is compensating the change in $\sigma_{\mathrm{iw}}$, which is due to the stronger impact of $f$ than $\Delta g^{\#}$ on $J_{\mathrm{imm}}$.

In case $\Delta g^{\#}$ is changed (only in Fig. 4b) the compensation is linear, following the structure of the nucleation rate formula.

Summarizing, an over/underestimation of $\sigma_{\mathrm{iw}}$ has a strong effect on the value of the resulting fit parameter, while an over/underestimation of $\Delta g^{\#}$ is less severe. If fit parameters were estimated based on fitting different CNT formulations they can not be directly compared. Fig. 4 can be used to estimate how different fit parameters would look like due to different



assumptions for $\sigma_{\mathrm{iw}}$ or $\Delta g^{\#}$. Some concrete examples/numbers how different fit parameter would look like if different CNT formulations are used for the fit are shown in Appendix B.

## 4    How to evaluate different CNT formulations?

The sensitivity analysis in section 3 raises the question how to evaluate different CNT formulations. Since most CNT formula-
tions with at least two fit parameters are able to reproduce the freezing curve of the measurements, it is not possible to use this reproducibility or goodness of the fit as the only measure for the physical capability of the used CNT formulation. In practice, mostly, contact angle schemes are judged based on the RMSE of the estimated and measured freezing curve. Only looking at the reproducibility of freezing curves however might be not conclusive enough since the fit parameters can mask uncertainties in the thermodynamic and kinetic parameters of CNT. Therefore, the evaluation of different CNT formulations has to be done
by stepwise testing different fit properties against measurements. The evaluation consists of a macroscopic and a microscopic perspective. Together, both perspectives yield three criteria.

At the macroscopic level: Is the CNT capable of reproducing the measured $FF$ at a given temperature and how well is the IN size and time dependence captured? The primary factor to test is the representation of temperature dependence, followed by the size of the IN and the predicted time dependence of the freezing process. All three dependencies should be captured by
a suitable CNT formulation if it is used as a function of $T$, $r_{\mathrm{IN}}$ and $t$ in a GCM.

When evaluating fits to measured freezing curves the goodness of the fit implicitly contains all three aspects (dependence on $T$, $r_{\mathrm{IN}}$ and $t$). However, because $T$ has the strongest effect on the $FF$, the goodness of fit mostly reflects how well the CNT scheme captures the temperature dependence.

Crit. 1  How accurately can the overall freezing curves be reproduced, i.e. how well is the temperature dependence of the $FF$
captured by the CNT formulation?

Crit. 2  How accurately are the particle size and time dependence of the freezing process captured by the CNT formulation? Criterion 2 can only be investigated if time and/or particle size dependent measurements are available.

At the microscopic level: Do the fit parameters match the microphysical assumptions of CNT, i.e. are the fit parameters physically reasonable? To evaluate if derived fit parameters are physically reasonable, the analysis of heterogeneous freezing
can be combined with the findings from homogeneous freezing. Including homogeneous freezing into the analysis might be useful because of less unconstrained parameters in this case.

Crit. 3  Are the values for the fit parameters reasonable in the context of what we know about the microphysical process of nucleation?

In the following these three criteria are used to decide which CNT formulations are suitable for parameterizing immersion
freezing, e.g. in a GCM.





## 5 Using experimental data to estimate CNT parameters for different contact angle schemes

In the following a comprehensive dataset of $FF$ (different aerosol species, aerosol particle sizes and residence times in the cloud chamber) is used. Five different mineral dust types were chosen for the analysis: Fluka kaolinite, illite-NX, montmorillonite, microcline (K-feldspar) and ATD (Arizona test dust). Montmorillonite or kaolinite are often used in global models as a

surrogate for ice nucleating dust, e.g. montmorillonite in ECHAM6-HAM2. They both represent clay minerals with kaolinite being a rather inefficient clay IN and montmorillonite an efficient clay IN. Fluka Kaolinite, which was used here, has been widely used to study the mechanism of immersion freezing. Illite-NX was chosen by the INUIT community as a the mineral dust reference sample, e.g. for instrument intercomparison (Hiranuma et al., 2014). Microcline (a sample from Namibia, variation Amazonit) and ATD were included to enable sensitivity studies of the freezing parameterization scheme with more

efficient IN. The experiments were done by A. Welti (Welti et al. (2012) and personal communication) using size-selected aerosol particles with diameters of 50, 100, 200, 400, 800 and 920 nm and 10 s residence time. Additional kaolinite measurements were done for different residence times of 1, 2, 3, 6, 9 and 21 s. Note that the residence times are rounded to full seconds [compared to Welti et al. (2012)] and not all datasets include the smallest and/or largest size (kaolinite: 100 - 920 nm, illite 100 - 800 nm, montmorillonite 100 - 800 nm, microcline 50 - 800 nm, ATD 100 - 800 nm). The error bars of the data reflect the

detection uncertainty and the statistical uncertainty in the measurement by multiple measurements.

To estimate the parameters of the CNT parameterization scheme four CNT formulations are chosen: #1 [single-$\alpha$ scheme], #2 [single-$\alpha$ scheme with $\Delta g^{\#}$ as a fit parameter], #5 [$\alpha$-pdf scheme] and #7 [$\alpha(T)$ scheme]. For more details see Table 2. Scheme #3, #4 and #6 use a $\sigma_{\mathrm{iw}}$, which was found not to represent homogeneous freezing well. The wrong assumption of $\sigma_{\mathrm{iw}}$ was chosen on purpose for the sensitivity study in section 3.1 to demonstrate how that influences the fit results and the freezing

curves. In the context of this section these formulations are excluded because they do not fulfill criterion 3. Note that also the single-$\alpha$ R&D + Z scheme (#1) is not expected to be able to reproduce the experimental freezing curves (criterion 1). It is still included here for comparison of the RMSE value with the other formulations ("bad" reference).

The fit parameters are determined by least square minimization of the calculated versus measured $FF$ from the dataset. For this purpose the dataset of each dust species, including all measurements as a function of $T$, aerosol particle size (diameter $d$)

and residence time ($t$), is used. To get an impression of the variability of the fit parameters throughout a dataset, the kaolinite dataset is additionally fitted for each size and time separately in Appendix C.

The fit parameters for the different CNT formulations and aerosol types are shown in Table 3 together with the best fit root mean square error (RMSE). The fit curves in comparison to the measured $FF$ are shown in Fig. 5 to Fig. 6 (in the case of kaolinite only a selection of the data is shown).

The geometric term $f$ in Table 3 is smallest for microcline, showing that this is the most efficient IN investigated here. The second lowest value for $f$ is found for ATD. Montmorillonite and kaolinite seem to be quite similar in terms of IN efficiency, whereas illite is the least efficient IN.

Revising the fit results with criterion 1 shows that scheme #1 is too steep and not able to reproduce experimental data, resulting in a high RMSE. One fit parameter is not enough to shift and reduce the steepness of the immersion freezing curve





sufficiently compared to homogeneous freezing. Thus the single-$\alpha$ scheme #1 does not fulfill criterion 1.

Reasonable fit results (low RMSE) are obtained with scheme #2 and #5 for all datasets. It is difficult to fit the data with scheme #7 since there is more than one solution for fit parameters (no absolute minimum of the fitting function). However, $\alpha_0$ should not become negative and $m$ has to be negative so that $\alpha$ increases with decreasing $T$. Otherwise criterion 3 is not fulfilled. Here

only the fit parameters that fullfill criterion 3 are given (local minimum of the fitting function).

Over all CNT formulations, the fits with largest RMSE are the ones for ATD which is probably caused by the mixed mineralogy of ATD. The capability of the different CNT formulations to best reproduce immersion freezing varies from dust to dust. Therefore, we establish a ranking for each dataset similar to the methodology of Wheeler et al. (2014). The best CNT formulation gets a ranking of 1, the worst a ranking of 4. From the ranking of the different datasets an average score is estimated

to judge the overall capability to predict $FF$ for each CNT formulation. The ranking (see Table 4) shows that scheme #7 and scheme #2 are the best followed by scheme #5 and scheme #1. Calculating the average RMSE from all fits (as an alternative) leads to a similar result, where the scheme #7 is the best and scheme #1) the worst (see also Table 4).

Note that this ranking does not consider criterion 3 and is only based on fit statistics. It also does not show directly how good the CNT formulations reproduce time and particle size dependence of the freezing process (criterion 2).

In section 6, the time and size dependence of the best three CNT formulations (scheme #2, scheme #5 and scheme #7) are compared to the kaolinite dataset. A deterministic immersion freezing parameterization scheme based on Niemand et al. (2012) is included in the evaluation for comparison (for more details see Appendix D). This scheme is frequently used in literature for comparing laboratory measurements, e.g. Atkinson et al. (2013); Hoffmann et al. (2013); Kanji et al. (2013); O'Sullivan et al. (2014); Tobo et al. (2014); Umo et al. (2015), but also as a parameterization scheme in some cases, e.g. Barahona et al. (2014);

Paukert and Hoose (2014); Hande et al. (2015).

## 6   Testing the time and particle size dependence (Criteria 2)

To test the ability of the single-$\alpha$ R&D + fit $\Delta g^{\#}$ scheme (#2), the $\alpha$-pdf R&D + Z scheme (#5) and the $\alpha(T)$ R&D + Z scheme (#7) to reproduce experimentally observed size and time dependence, the fit parameters for kaolinite (see Table 3) are used to calculate $FF$s for three different residence times (1, 10 and 21 s) and three different aerosol diameters (100, 400, 800 nm). In

the case of the size dependent calculation of $FF$ the time is 10 s, in the case of the time dependent calculation the diameter is 400 nm. The calculated $FF$ is compared to measurements of the size and time dependent $FF$ in Fig. 5. The analysis of the RMSE for each dataset and CNT formulation revealed marginal differences in the second decimal place and is therefore not shown.

Figure 5 shows that scheme #5 is able to represent the time and particle size dependence better than scheme #2 and #7.

This leads to an overall smaller RMSE and explains the better ranking for scheme #5 in the case of kaolinite (see Table 3). Looking at Fig. 5, the scheme #2 and #7 seem to overpredict both the size and time dependence, while scheme #5 seems to underpredict the particle size dependence but captures the time dependence well. Overpredicting the size dependence translates into an overestimation of $FF$ for particles with an aerosol particle diameter larger than 400 nm and an underestimation of $FF$





for particles with an aerosol particle diameter smaller than 400 nm. Underpredicting the size dependence has the opposite influence on $FF$. Overpredicting the time dependence means that $FF$ is overestimated having a larger timestep (larger 10 s) as in GCMs.

Note, that the outcome of the evaluation depends on the dataset used. For different aerosol species the ranking of scheme #5
and scheme #2 differs, e.g. for montmorillonite, microcline and ATD. Due to this limitation, it cannot generally be concluded which contact angle scheme better fulfills criterion 2. Since all three schemes are computationally equally expensive (if the contact angle distribution is not changed with time), they all might be chosen for CNT based immersion freezing parameterization schemes in GCMs.

In Fig. 5 the CNT curves are also compared to an empirical immersion freezing parameterization scheme based on the ex-
pression given in Niemand et al. (2012). Since the scheme is dependent on the measurement data used to derive it, the use for comparison is only limited. To be able to compare an empirical scheme to the CNT schemes, a $n_{s,IN}$ scheme similar to the one in Niemand et al. (2012) was fitted to kaolinite measurements. Details can be found in Appendix D. The $n_{s,IN}$ scheme slightly overestimates the particle size dependence and the scheme does not capture any time dependence since it is deterministic. It is able to represent freezing curves in a similar manner as CNT. However, due to the general characteristics of empirical relations
it is not clear if it can be extrapolated to a wide $T$-range, which would be mandatory for the use in a GCM.

## 7   Conclusions

In this study the sensitivity of CNT based immersion freezing parameterization schemes to thermodynamic and kinetic parameters is investigated as a response to the large sensitivity of homogeneous freezing on these parameters. For the use in models a validation of these sensitivities is important to estimate uncertainties coming from different parameterization schemes which
include the effect of aerosol particles on the energy barrier of ice nucleation.

Compared to homogeneous freezing, immersion freezing has one more unconstrained parameter, namely the contact angle $\alpha$. Different schemes to represent the contact angle/-distribution with one or two fit parameters based on experimental data are tested. It is found that several different contact angle schemes are able to reproduce the experimental freezing curve. These schemes contain two fit parameter, while the contact angle scheme with only one fit parameter, the single-$\alpha$ scheme, cannot
reproduce the freezing curves.

Analyzing the importance of the choice of $\sigma_{iw}$ and $\Delta g^{\#}$ to parameterize immersion freezing revealed that uncertainties in the thermodynamics or kinetics can be compensated by two-parameter contact angle schemes. As a result an under/overestimation of $\sigma_{iw}/\Delta g^{\#}$ does not lead to a bad representation of freezing curves as in case of homogeneous freezing. Because the fit parameters compensate inaccuracies or uncertainties of the thermodynamic and kinetic parameters, the absolute value of the
found fit parameters is highly dependent on the choice of thermodynamic/kinetic parameters within the formulation of CNT (especially on $\sigma_{iw}$). As a consequence, contact angles for CNT parameterization schemes from different authors can only be applied within the same CNT formulation. Implementing one formulation of CNT into a GCM together with a different estimate for $\alpha$ might introduce an offset into modeling studies (see red curve in Fig. 3). Besides the sensitivity of $\alpha$ on the





thermodynamic and kinetic parameters used in CNT makes a direct comparison of contact angle values derived in different studies impossible. It again stresses the importance of highlighting which CNT formulation was used for the analysis of experimental data [as stressed in Ickes et al. (2015)].

Another consequence is that the reproducibility of freezing curves should not be the only criterion to decide on a CNT
formulation because it can be misleading if the formulation has at least two fit parameters. Therefore some criteria to evaluate reasonable CNT formulations are compiled here. They additionally take into account the microphysical perspective (assumptions the CNT formulation is based on) and the ability to predict the size and time dependence of the freezing process. Particle size and nucleation time are implicitly included in the reproducibility of freezing curves but should be evaluated separately.

The fit parameters for scheme #2, scheme #5 and scheme #7 are determined for five different datasets of mineral dust
(kaolinite Fluka, NX illite, montmorillonite, microcline and ATD) by fitting the CNT approach to the $FF$ from measurements. Good results in reproducing the freezing curves (criterion 1: $T$-dependence and partly 2: size and time dependence) are achieved when using a single-$\alpha$ scheme with fitted constant $\Delta g^{\#}$ (CNT #2), an $\alpha$-pdf scheme (CNT #5) or an $\alpha(T)$ scheme (CNT #7) when ignoring unreasonable solutions for the fit parameters (criterion 3). The single-$\alpha$ scheme does not perform well when $\Delta g^{\#}$ is not used as an additional fit parameter.

The three good working CNT formulations (#2, #5 and #7) are further evaluated by looking how well they reproduce the time and particle size dependence of the kaolinite dataset (criterion 2). In this case the $\alpha$-pdf scheme (CNT #5) works better, as it captures the time dependence. However, the particle size dependence is underpredicted. Using a single-$\alpha$ scheme with fitted constant $\Delta g^{\#}$ (CNT #2) or an $\alpha(T)$ scheme (CNT #7) overpredicts the size and time dependence. Note that the results only refer to the kaolinite dataset. Due to this restriction it remains ambiguous which CNT formulation best fulfills criterion 2 and
thus is best suited for modeling purpose. It would be helpful to redo the analysis for other dust types. From the perspective of criterion 2 all three CNT formulations seem to be able to predict nucleation rates for mineral dust particles.

An empirical immersion freezing parameterization scheme [based on Niemand et al. (2012)] can also capture the freezing curves and IN size dependence quite well. However, it is not clear if it is legitimate to extrapolate the empirical relationship, so that the full $T$-range is covered.

Criterion 3 is difficult to evaluate coming from a macroscopic level as microphysical knowledge is missing at this point. Scheme #5 is consistent with the microphysical perspective of freezing. Evaluating scheme #2 requires knowledge about a possible influence on $\Delta g^{\#}$ by an aerosol particle immersed in the supercooled droplet and is thus not possible. Scheme #7 can return unphysical fit parameters (criterion 3 is not always fulfilled). However, this evaluation is limited by the recognition how $\alpha$ might change with temperature.

More size and time dependent measurements of different IN would be beneficial to evaluate different CNT formulations more robustly.



**Table 1.** List of symbols

| Symbol | Unit | Description |
|---|---|---|
| $A_{\mathrm{IN}}$ | $\mathrm{m}^2$ | Surface area of an IN |
| $A_{\mathrm{tot}},$ | $\mathrm{m}^{-2}$ | Total surface area per unit volume of particles over all size bins |
| $A_{\mathrm{tot},j}$ | $\mathrm{m}^{-2}$ | Total surface area per unit volume of particles in size bin $j$ |
| $A_j$ | $\mathrm{m}^{-2}$ | Dust particle surface area in size bin $j$ |
| $C_{\mathrm{prefac,hom}}$ | $\mathrm{m}^{-3}\,\mathrm{s}^{-1}$ | Preexponential factor of the homogeneous nucleation rate |
| $d$ | m | Aerosol particle diameter |
| $f$ | - | Geometric term |
| $h$ | J s | Planck constant |
| $J_{\mathrm{hom}}$ | $\mathrm{m}^{-3}\,\mathrm{s}^{-1}$ | Homogeneous nucleation rate |
| $J_{\mathrm{imm}}$ | $\mathrm{m}^{-2}\,\mathrm{s}^{-1}$ | Immersion freezing nucleation rate |
| $k_{\mathrm{B}}$ | $\mathrm{J\,K}^{-1}$ | Boltzmann constant |
| $n_{k,\mathrm{germ}}$ | - | Number of water molecules in the ice germ |
| $n_{\mathrm{s}}$ | $\mathrm{m}^{-2}$ | Number of water molecules in contact with the unit area of the ice germ |
| $n_{\mathrm{s,IN}}$ | $\mathrm{m}^{-2}$ | Surface density of active sites on an IN |
| $N_{\mathrm{i}}$ | - | Ice crystal number concentration |
| $N_{\mathrm{i},j}$ | - | Number of ice active aerosol particles in size bin $j$ |
| $N_{\mathrm{l}}$ | $\mathrm{m}^{-3}$ | Volume number density of water molecule in liquid water |
| $N_{\mathrm{tot},j}$ | - | Total number of aerosol particles in the size bin $j$ |
| $m$ | $\mathrm{rad\,K}^{-1}$ | Change of contact angle with temperature |
| $p(\alpha)$ | - | Probability density of contact angle $\alpha$ |
| $r_{\mathrm{germ}}$ | m | Radius of the ice germ (=critical radius) |
| $r_{\mathrm{IN}}$ | m | Radius of the IN |
| $S_{\mathrm{i}}$ | - | Saturation ratio with respect to ice |
| $t$ | s | Time |
| $T$ | K | Temperature |
| $v_{\mathrm{ice}}$ | $\mathrm{m}^3$ | Volume of a water molecule in the ice embryo |
| $Z$ | - | Zeldovich factor |
| $\alpha$ | rad | Contact angle |
| $\alpha_0$ | rad | Contact angle at melting point |
| $\Delta g^{\#}$ | J | Activation energy barrier |
| $\Delta G, \Delta G_{\mathrm{hom}}$ | J | Gibbs free energy barrier (of homogeneous freezing) |
| $\Delta t$ | s | Time step |
| $\mu$ | rad | Mean contact angle of the contact angle distribution |
| $\sigma$ | - | Variance of the contact angle distribution |
| $\sigma_{\mathrm{iw}}$ | $\mathrm{J\,m}^{-2}$ | Interfacial tension between ice/water |
| $\pi$ | - | Ratio of a circle's circumference to its diameter |





**Table 2.** Overview of CNT formulations used for the sensitivity analysis, results for the fit parameters and evaluation of the fit result. The values are rounded to two digits after the decimal point.

CNT formulation #1, #5 and #7 use the best fitting combination of $\sigma_{\text{iw}}$ and $\Delta g^{\#}$ emerging from the homogeneous freezing analysis in Ickes et al. (2015).

| # | Formulation for $\sigma_{\text{iw}}$ and $\Delta g^{\#}$ | Contact angle scheme | Name | Fit parameters | RMSE |
|---|---|---|---|---|---|
| 1 | $\sigma_{\text{iw}}$: Reinhardt and Doye (2013) $\Delta g^{\#}$: Zobrist et al. (2007)$\approx 5 \cdot 10^{-20}$J | single-$\alpha$ | Single-$\alpha$ R&D + Z | 1: $f$=0.55 | 0.12 |
| 2 | $\sigma_{\text{iw}}$: Reinhardt and Doye (2013) $\Delta g^{\#}$: constant | single-$\alpha$ | Single-$\alpha$ R&D + fit $\Delta g^{\#}$ | 2: $f$=0.24 $\Delta g^{\#}$=11.01$\cdot 10^{-20}$J | 0.05 |
| 3 | $\sigma_{\text{iw}}$ from Ouchi (1954) $\Delta g^{\#}$: constant | single-$\alpha$ | Single-$\alpha$ O + fit $\Delta g^{\#}$ | 2: $f$=0.69 $\Delta g^{\#}$=12.3$\cdot 10^{-20}$J | 0.05 |
| 4 | $\sigma_{\text{iw}}$ from Eadie (1971) $\Delta g^{\#}$: constant | single-$\alpha$ | Single-$\alpha$ E + fit $\Delta g^{\#}$ | 2: $f$=0.23 $\Delta g^{\#}$=10.46$\cdot 10^{-20}$J | 0.05 |
| 2/4 | $\sigma_{\text{iw}}$: Reinhardt and Doye (2013) $\Delta g^{\#}$: constant | single-$\alpha$ | Single-$\alpha$ R&D + fit $\Delta g^{\#}$(E) | 2: $f$=0.24 (#2) $\Delta g^{\#}$=10.46$\cdot 10^{-20}$J (#4) | 0.31 |
| 5 | same as #1 | $\alpha$-pdf | $\alpha$-pdf R&D + Z | 2: $\mu$=0.5 rad $\sigma$=0.04 | 0.05 |
| 6 | $\sigma_{\text{iw}}$ from Eadie (1971) $\Delta g^{\#}$: Zobrist et al. (2007) | $\alpha$-pdf | $\alpha$-pdf E + Z | 2: $\mu$=0.44 rad $\sigma$=0.03 | 0.05 |
| 7 | same as #1 | $\alpha(T)$ | $\alpha(T)$ R&D + Z | 2: $\alpha_0$=0.7 $m$= -0.03 | 0.05 |





**Table 3.** Estimated fit parameters for the different CNT formulations used for different mineral dust types. The values are rounded to two digits after the decimal point.

| # | Fit parameter | Kao | RMSE | Ill | RMSE | Mont | RMSE | Micro | RMSE |
|---|---|---|---|---|---|---|---|---|---|
| 1 | $f$ | 0.56 | 0.2 | 0.61 | 0.17 | 0.56 | 0.18 | 0.3 | 0.22 |
| 2 | $f$ | 0.29 | 0.14 | 0.36 | 0.14 | 0.28 | 0.09 | 0.11 | 0.1 |
|   | $\Delta g^{\#}/10^{-20}$ J | 9.95 | | 8.93 | | 10.03 | | 11.97 | |
| 5 | $\mu$/rad | 0.5 | 0.09 | 0.54 | 0.13 | 0.5 | 0.15 | 0.25 | 0.13 |
|   | $\sigma$ | 0.06 | | 0.05 | | 0.04 | | 0.11 | |
| 7 | $\alpha_0$/rad | 0.84 | 0.14 | 0.98 | 0.13 | 0.81 | 0.09 | 0.61 | 0.1 |
|   | $m$ | -0.02 | | -0.02 | | -0.02 | | -0.03 | |

| # | Fit parameter | ATD | RMSE |
|---|---|---|---|
| 1 | $f$ | 0.58 | 0.32 |
| 2 | $f$ | 0.14 | 0.21 |
|   | $\Delta g^{\#}/10^{-20}$ J | 12.58 | |
| 5 | $\mu$/rad | 0.48 | 0.25 |
|   | $\sigma$ | 0.16 | |
| 7 | $\alpha_0$/rad | 0.39 | 0.13 |
|   | $m$ | -0.04 | |

**Table 4.** Ranking of the capability of the different CNT formulations to reproduce the freezing curves for different mineral dust particles.

| # | Kaolinite | Illite | Montmorillonite | Microcline | ATD | Average score |
|---|---|---|---|---|---|---|
| 1 | 3 | 3 | 3 | 3 | 4 | 3.2 |
| 2 | 2 | 2 | 1 | 1 | 2 | 1.6 |
| 5 | 1 | 1 | 2 | 2 | 3 | 1.8 |
| 7 | 2 | 1 | 1 | 1 | 1 | 1.2 |

| # | Average RMSE | Score based on average RMSE |
|---|---|---|
| 1 | 0.218 | 4 |
| 2 | 0.136 | 2 |
| 5 | 0.15 | 3 |
| 7 | 0.118 | 1 |





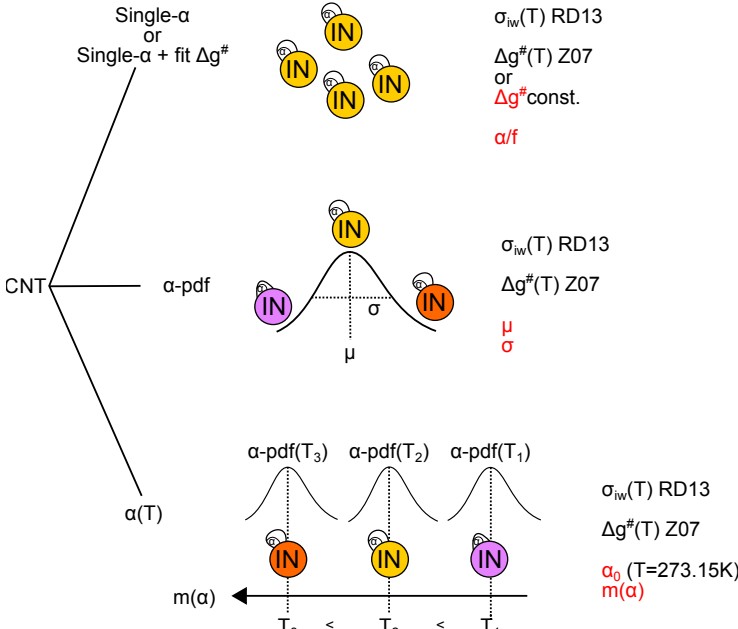

**Figure 1.** Schematic sketch of different contact angle schemes.

**Stochastic:** all IN contain the same contact angle $\alpha$ (single-$\alpha$). The ice germ including the contact angle $\alpha$ is shown.

**Semi-singular:** IN contain different contact angle $\alpha$, but each IN has a single $\alpha$. The contact angles are distributed with an $\alpha$-pdf over the IN population. The sketch shows three different contact angles $\alpha_1$ (orange), $\alpha_2$ (yellow) and $\alpha_3$ (purple) of the contact angle distribution. The least efficient contact angle $\alpha_1$ is the largest ($> \alpha_2 > \alpha_3$).

**Simplified semi-singular:** IN contain a singular contact angle with $\alpha_1 > \alpha_2 > \alpha_3$ shown as an example. The contact angle is equivalent to mean contact angle $\mu$ of the $\alpha$-pdf scheme, which changes with temperature in analogon to the time/temperature evolution of the $\alpha$-pdf [$\alpha$(T)]. The fit parameters for each contact angle scheme are marked in red. Figure adapted from F. Lüönd.





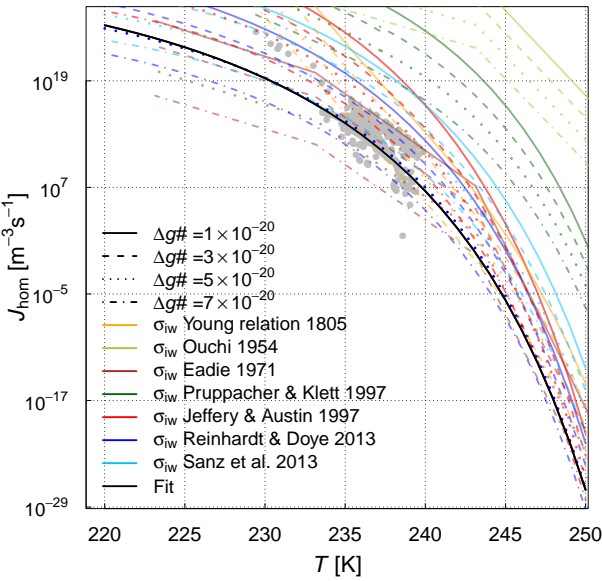

**Figure 2.** Comparison of the fitted $J_{\mathrm{hom}}(T)$ (solid black line) with calculated nucleation rates using different formulations of $\sigma_{\mathrm{iw}}$ and constant values for $\Delta g^{\#}$. Grey dots show the collected homogeneous freezing dataset. $\sigma_{\mathrm{iw}}$ from Reinhardt and Doye (2013) captures the homogeneous freezing curve the best.

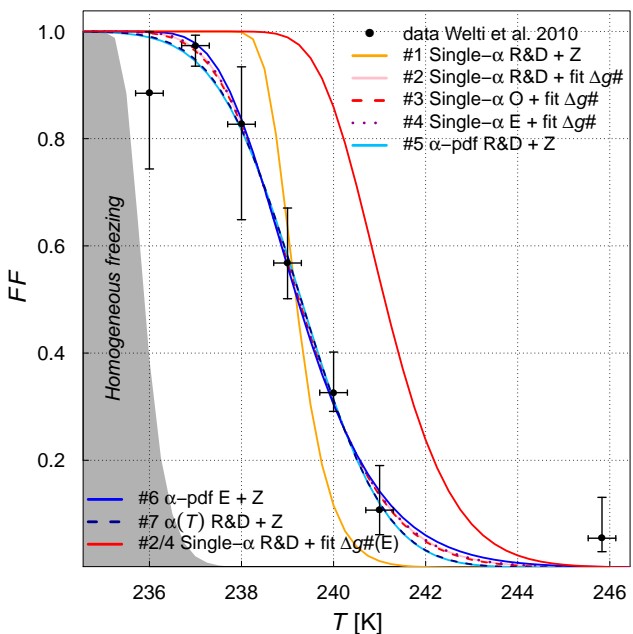

**Figure 3.** Calculated frozen fraction $FF$ as a function of $T$ for different thermodynamic and kinetic parameters in combination with different contact angle schemes for kaolinite with a particle diameter of 400 nm after a residence time of 10 s. More details can be found in Table 2.





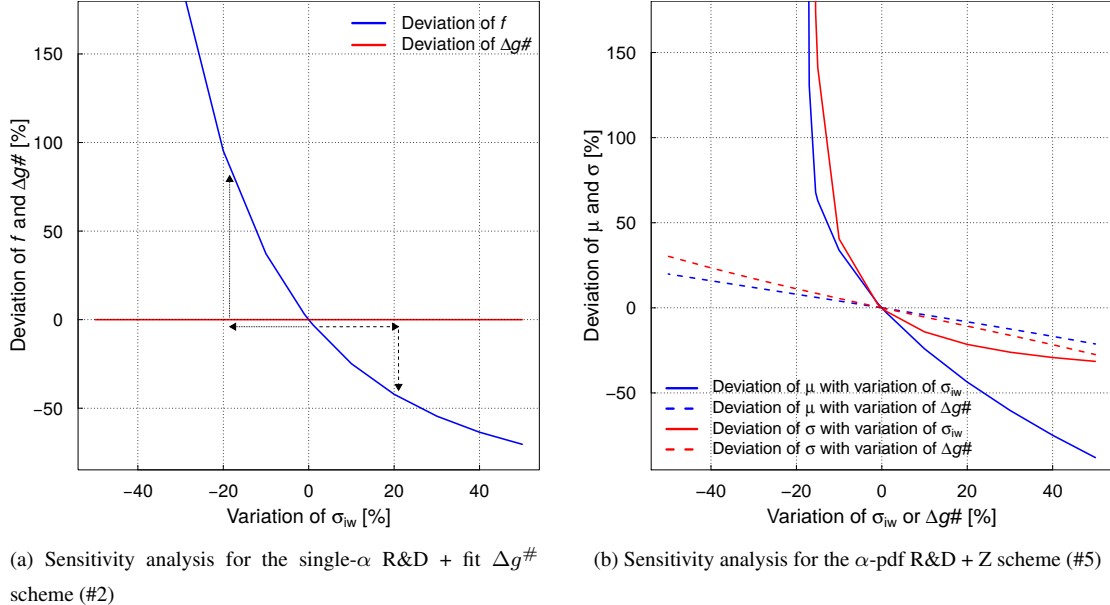

(a) Sensitivity analysis for the single-$\alpha$ R&D + fit $\Delta g^{\#}$ scheme (#2)

(b) Sensitivity analysis for the $\alpha$-pdf R&D + Z scheme (#5)

**Figure 4.** Magnitude of deviation from a reference fit in percent (relative uncertainty). The results are shown for a variation of $\sigma_{iw}$ and $\Delta g^{\#}$ from 0 to $\pm 50\%$. The applied change is indicated by the line type (solid = change in thermodynamics, dashed = change in kinetics). The figure can be used to estimate the direction in which fit parameters deviate.







**Figure 5.** Calculated $FF$ of kaolinite for certain times and sizes using the single-$\alpha$ R&D + fit $\Delta g^{\#}$ scheme (#2), the $\alpha$-pdf R&D + Z scheme (#5) and the $\alpha(T)$ R&D + Z scheme (#7) with corresponding fit parameters (see Table 3) and a simplified immersion freezing parameterization scheme based on Niemand et al. (2012) compared to the dataset. Figure (a)-(c) show the particle size dependence (t=10 s), Fig. (c)-(e) show the time dependence (d=400 nm).





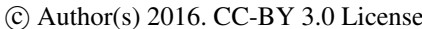

**Figure 6.** Calculated $FF$ of illite, montmorillonite, microcline and ATD after a residence time $t$ of 10 s for certain sizes using the single-$\alpha$ R&D + fit $\Delta g^{\#}$ scheme (#2), the $\alpha$-pdf R&D + Z scheme (#5) and the $\alpha(T)$ R&D + Z scheme (#7) with corresponding fit parameters (see Table 3) and a simplified immersion freezing parameterization scheme based on Niemand et al. (2012) compared to the dataset.





(a) Montmorillonite, d=400 nm

(b) Montmorillonite, d=800 nm

(c) Microcline, d=50 nm

(d) Microcline, d=100 nm

(e) Microcline, d=200 nm

(f) Microcline, d=400 nm

**Figure 6.** Continued.





(a) Microcline, d=800 nm

(b) ATD, d=100 nm, t=10 s

(c) ATD, d=200 nm, t=10 s

(d) ATD, d=400 nm, t=10 s

(e) ATD, d=800 nm, t=10 s

**Figure 6.** Continued.





# Appendix A: Analysis of the different contact angle scheme formulations: How do the fit parameters influence the calculated nucleation rate/frozen fraction

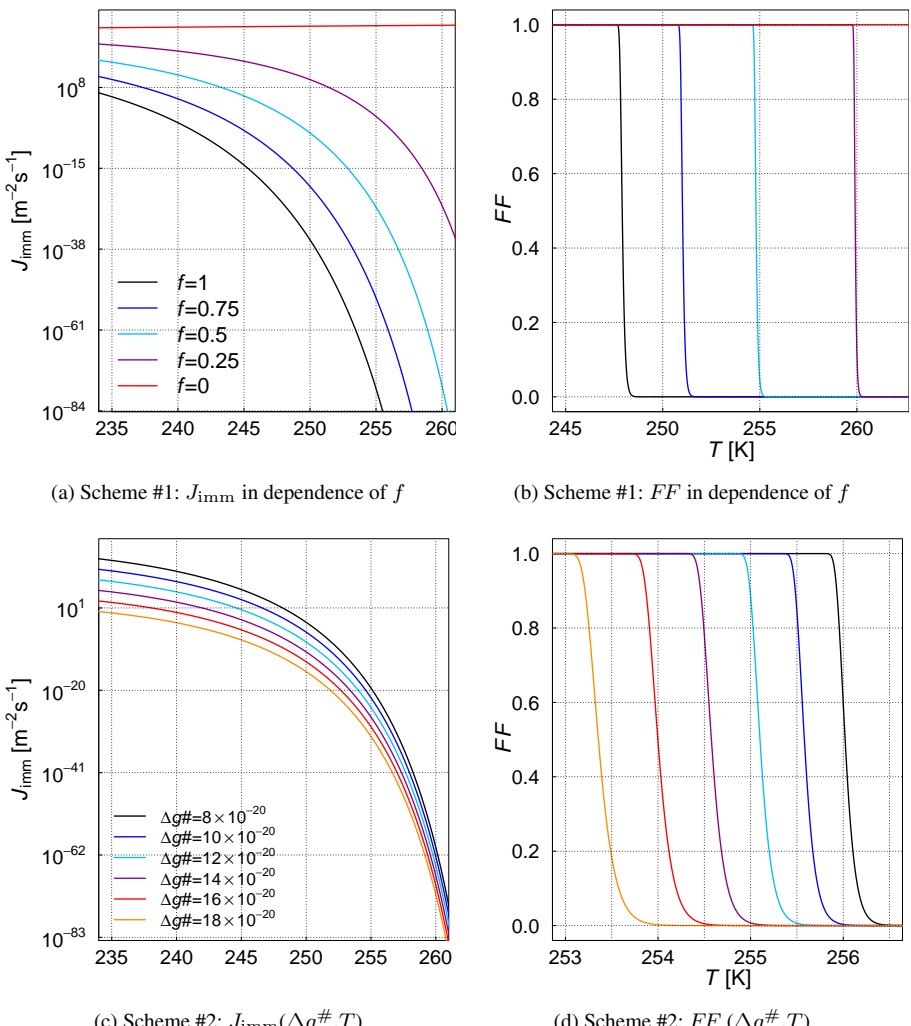

(a) Scheme #1: $J_{\mathrm{imm}}$ in dependence of $f$

(b) Scheme #1: $FF$ in dependence of $f$

(c) Scheme #2: $J_{\mathrm{imm}}(\Delta g^{\#}, T)$

(d) Scheme #2: $FF (\Delta g^{\#}, T)$

**Figure 7.** Nucleation rate $J_{\mathrm{imm}}$ and frozen fraction $FF$ in dependence of the fit parameters for the different contact-angle schemes (here $f$ and $\Delta g^{\#}$).

In case of the single-$\alpha$ R&D + fit $\Delta g^{\#}$ scheme (#2) the geometric term $f$ was chosen to be 0.4. Note that the dependence of the scheme (in the case of a fixed $\Delta g^{\#}$) on $f$ is the same as with the single-$\alpha$ R&D + Z scheme (#1).

Decreasing $f$ (reducing the energy barrier) shifts the freezing curve to warmer temperatures. The slope changes only negligible. Increasing $\Delta g^{\#}$ (increasing the activation energy barrier) shifts the freezing curve to lower temperatures. It also changes the slope of the curve- a higher activation energy barrier leads to a flattening of the curve.




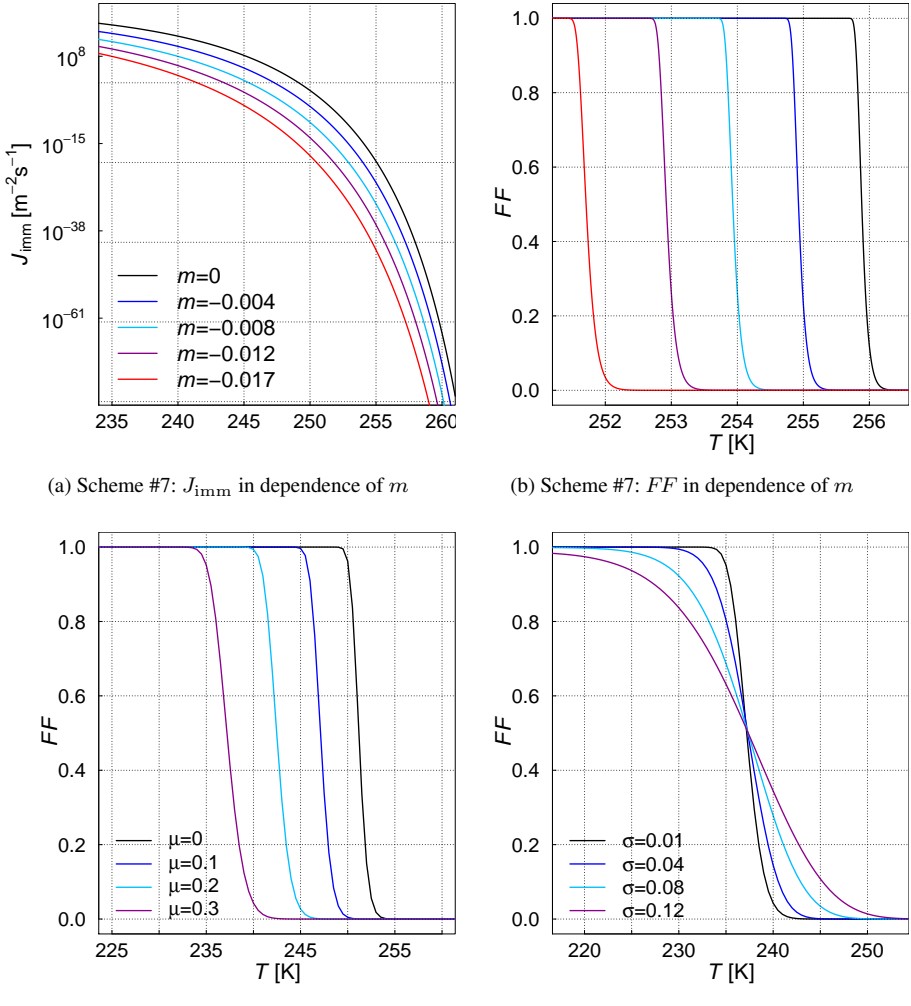

(a) Scheme #7: $J_{\mathrm{imm}}$ in dependence of $m$

(b) Scheme #7: $FF$ in dependence of $m$

(c) Scheme #5: $FF$ in dependence of $\mu$

(d) Scheme #5: $FF$ in dependence of $\sigma$

**Figure 8.** Nucleation rate $J_{\mathrm{imm}}$ and frozen fraction $FF$ in dependence of the fit parameters for the different contact-angle schemes (here $m$, $\mu$ and $\sigma$).

In case of scheme #7 $\alpha_0$ was chosen to be 1.5 ($\approx 85°$). Note that the dependence of the scheme (in the case of a fixed change in contact angle $m$) on $\alpha_0$ is the same as the change with $f$ of the single-$\alpha$ R&D + Z scheme (#1).

In case of the $\alpha$-pdf R&D + Z scheme (#5) $\sigma$ was chosen to be 0.01 in the left figure, $\mu$ was chosen to be 0.3 ($\approx 18°$) in the right figure.

Decreasing $m$ (contact angle gets larger with $T$ because most efficient IN are used first) shifts the freezing curve to colder temperatures and changes the slope of the curve (flattening).

Increasing $\mu$ (increasing the contact angle) shifts the freezing curve to lower temperatures and flattens the curve, while an increase in $\sigma$ (broadening of the contact angle distribution) changes the slope of the freezing curve only (flattening). Note that already small changes in $\mu$ lead to a considerable shift of the curve compared to the other schemes.



**Appendix B: Estimating the relative uncertainty of fitting $\alpha$: Example calculations**

Figure 4 can be used to estimate the deviation of fit parameters from different CNT formulations relative to each other. To show this we estimate the difference in fit parameter when using $\sigma_{iw}$ from Eadie (1971) instead of Reinhardt and Doye (2013) in combination with an $\alpha$-pdf scheme [scheme (#6) compared to scheme (#5)].

5  Within the 10 K temperature range of the immersion freezing measurements (236-246 K) $\sigma_{iw}$ is on average 4% higher when using $\sigma_{iw}$ from Eadie (1971) instead of Reinhardt and Doye (2013). An increase in $\sigma_{iw}$ by 2.5% (246 K) or 5% (236 K) would lead to a decrease in $\mu$ by approximately 7 to 13% (see Fig. 4 b). Now we check if that estimated change matches with the real change when fitting the same dataset with the two different $\sigma_{iw}$. In Table 2 using $\sigma_{iw}$ from Eadie (1971) leads to a mean contact angle of 0.44 rad (approx. 25.5°) instead of 0.5 rad (approx. 28°) when using $\sigma_{iw}$ from Reinhardt and Doye (2013). This is a

10  difference of 12%, conform with the estimate from Fig. 4 (approx. 7-13%). However, the variance $\sigma$ of the $\alpha$-pdf distribution is expected to change less (5 to 9%) but a change by 25% is found.

In some cases the predicted change in fit parameters from Fig. 4 deviates from the real change in fit parameters (Table 2). The problem with Fig. 4 is, that the assumption of a constant variation of $\sigma_{iw}$ is invalid in most cases, so that Fig. 4 can not be used easily. However, it can be used to illustrate how fit results might change and estimate a rough deviation from the

15  reference when using different thermodynamic and kinetic parameters especially for cases where $\sigma_{iw}$ changes nearly constant over the fitted temperature range. This can help when comparing fit results to fit results from another study where a different formulation of CNT was used.





## Appendix C:  Variability of the fit parameters throughout one dataset

The variability of the fit parameters throughout the dataset can be seen when fitting the single-$\alpha$ R&D + fit $\Delta g^{\#}$ scheme (#2) and the $\alpha$-pdf R&D + Z scheme (#5) to $FF$ of kaolinite data for different sizes and residence times separately. The resulting fit parameters are compared in Fig. 10 for different sizes in red and for different times in blue and light blue. Each point in Fig. 10 represents the value of the best fit parameter for one subset of the kaolinite dataset. The labels on the x-axis give information which subset of the dataset was fitted. The residence time is 10 s for the data subsets of different sizes and the diameter of the kaolinite particles 400 nm or 800 nm for the data subsets of different times (blue/light blue). The dashed line indicates the mean of the size or time dependent fit parameter. The standard deviation is shown as shaded box.

The fit parameters vary depending on the measurement conditions. Omitting the measurement with the smallest aerosol particle size (d=100 nm) and the shortest residence time (1 s) the variation between the data subsets is small. The variability of the fit parameters is larger for different aerosol particle sizes compared to different residence times, which might be due to the higher sensitivity of the freezing process to particle size compared to time. For scheme #2 the fit parameters seem to be correlated. High values of one fit parameter, e.g. $f$, correspond to low values of the other fit parameter, e.g. $\Delta g^{\#}$. Scheme #5 on the other hand does not show a clear correlation.

The different fit results for scheme #5 can be used to study how the shape of the contact angle distribution might change with the size of the particles or the residence time.

Whereas the fitted contact angle distribution does not change noticeable with time between 1 and 21 s (Figure not shown here), the variance $\sigma$ changes with particle size (see Fig. 9). The contact angle distribution broadens with increasing aerosol particle size (neglecting the fit of the 400 nm dataset, which does not fit into the picture), which reflects a larger probability of different $\alpha$ on the aerosol particle population with increasing size. The particle population is more heterogeneous. Additionally the maximum is shifting to the left (smaller contact angle) which means that the IN is getting more efficient with size. Note that the curves are not considering measurement uncertainties of the fitted data and therefore can only used to qualitatively interpret the result. In case of idealized measurements the result could be used to derive a relationship for the width of the contact angle distribution and the size of the IN.

## C1   Uncertainty of fit parameters due to limited data

In many cases there are not size- and time-dependent measurements available. Here we investigate the quality of fit parameters if only limited amount of data is available. For that purpose we use the kaolinite dataset (as this is the most thoroughly dataset available within this study) and use only subsets of the dataset assuming that not all data is available to estimate the fit parameters. The quality of the gained fit parameters is then estimated by using the complete dataset and look how good the freezing curves can be represented (RMSE). We look at four different cases:

1. Reference, the whole dataset is fitted (see also Table 3).

2. Only size dependent measurements are available, time dependence is not known ($t$=10 s).





3. Only time dependent measurements are available, size dependence is not known ($d$=400 nm).

4. Only one measurement is available ($t$=10 s, $d$=400 nm).

The resulting fit parameters and the deviation from the kaolinite dataset is shown in Table 5. The fit parameters are not significantly different when the dataset is limited to only size or only time dependent data. Also the deviation from the complete

5   dataset is not significant (RMSE). This analysis therefore does not allow any conclusion how many dependencies, e.g. size and time, have to be taken into account to successfully fit freezing curves. However if using only one single dataset, the results for the fit parameters are different and the deviation from the measurements is higher. Note that the deviation when fitting only a single dataset could be larger if a dataset is chosen which is not similar to the average values of the dataset as in this case. This means that there is no guarantee that fits can be extrapolated/used in a universal way across different conditions.

**Table 5.** Estimated fit parameters for the different CNT formulations used for kaolinite using the complete dataset or subsets of the dataset. The values are rounded to two digits after the decimal point. The RMSE value shows the deviation of the fit to the complete dataset.

| # | Fit parameter | Reference | RMSE | Only size | RMSE | Only time | RMSE | Only one dataset | RMSE |
|---|---|---|---|---|---|---|---|---|---|
| 2 | $f$ | 0.29 | 0.14 | 0.29 | 0.14 | 0.3 | 0.14 | 0.23 | 0.15 |
|   | $\Delta g^{\#}/10^{-20}$ J | 9.95 | | 9.93 | | 9.7 | | 11.01 | |
| 5 | $\mu$/rad | 0.5 | 0.09 | 0.5 | 0.1 | 0.5 | 0.09 | 0.49 | 0.12 |
|   | $\sigma$ | 0.06 | | 0.05 | | 0.07 | | 0.04 | |

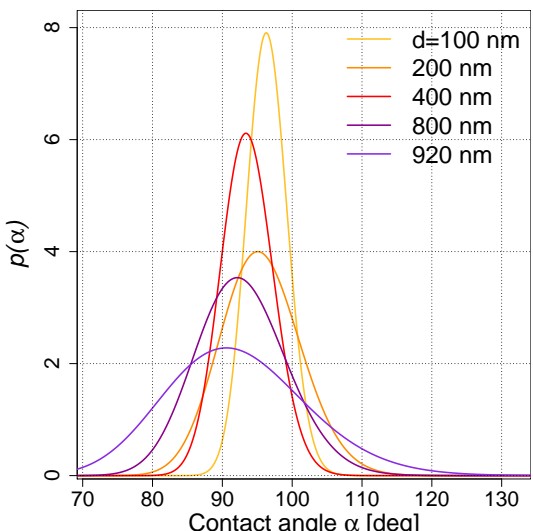

**Figure 9.** Change of $\alpha$-pdf with particle size for the kaolinite dataset. The residence time is 10 s.





**Figure 10.** Variability of the fit parameters throughout the kaolinite dataset. The variability with the aerosol particle size ($d$) is shown for a residence time $t$ of 10 s (red). The variability with the time ($t_1$ and $t_2$) is shown for a aerosol particle diameter of 400 nm ($t_1$, blue) and 800 nm ($t_2$, lightblue). The first row shows the fit results for scheme #2, the second row for scheme #5. The RMSE value shows the deviation of the fit to the single dataset it was fitted to.



## Appendix D: Estimating the surface site density $n_{\mathrm{s,IN}}$

The Niemand et al. (2012) scheme for immersion freezing of natural dust, is a deterministic scheme based on the approach of Connolly et al. (2009). It is derived from measurements carried out at the "Aerosol Interaction and Dynamics in the Atmosphere" (AIDA) cloud chamber at KIT, Karlsruhe. A detailed description of the cloud chamber and the measurements can be found in Niemand et al. (2012).

In this approach it is assumed that ice nucleation is a function of $T$ and particle surface area $A_{\mathrm{IN}}$ because the amount of active sites scales with particle size.

In Connolly et al. (2009) the change of the number of ice active aerosol particles in the size bin $j$, $N_{\mathrm{i},j}$, with respect to $T$ is:

$$\frac{dN_{\mathrm{i},j}}{dT} = (N_{\mathrm{i},j} - N_{\mathrm{tot},j}) \cdot A_j \cdot k(T) , \tag{D1}$$

where $N_{\mathrm{tot},j}$ denotes the total number of aerosol particles in the size bin $j$ and $A_j$ for the dust particle surface area in the same size bin. The surface site density of ice active sites $n_{\mathrm{s,IN}}$ as a function of $T$ can be determined by integrating the factor $k(T)$ over the whole temperature range:

$$n_{\mathrm{s,IN}}(T) = \int_{T}^{0} k(T) dT . \tag{D2}$$

Using Eq. D1 and D2 the frozen fraction $FF$ can be expressed as a function of $T$:

$$FF = \frac{N_{\mathrm{i},j}}{N_{\mathrm{tot},j}} = 1 - \exp(-A_j \cdot n_{\mathrm{s,IN}}(T)) \approx A_j \cdot n_{\mathrm{s,IN}}(T) . \tag{D3}$$

The approximation is valid for $A_j \cdot n_{\mathrm{s,IN}}(T) \ll 1$, which translates into small particles and high temperatures. For low temperatures, e.g. 243.15 K and particles larger than 3 $\mu$m the term $A_j \cdot n_{\mathrm{s,IN}}$ is approximately 1.

The surface site density of ice active sites, $n_{\mathrm{s,IN}}(T)$, is calculated from the total surface area of aerosol particles in the AIDA chamber and the measured ice crystal number concentration during one freezing experiment:

$$\sum_{j=1}^{n} N_{\mathrm{i},j} \quad \approx \quad \sum_{j=1}^{n} N_{\mathrm{tot},j} \cdot A_j \cdot n_{\mathrm{s,IN}}(T) = n_{\mathrm{s,IN}}(T) \cdot \sum_{j=1}^{n} N_{\mathrm{tot},j} \cdot A_j \tag{D4}$$

$$\Leftrightarrow n_{\mathrm{s,IN}}(T) \quad = \quad \frac{\sum_{j=1}^{n} N_{\mathrm{i},j}}{\sum_{j=1}^{n} N_{\mathrm{tot},j} \cdot A_j} = \frac{\sum_{j=1}^{n} N_{\mathrm{i},j}}{\sum_{j=1}^{n} A_{\mathrm{tot},j}} = \frac{N_{\mathrm{i}}}{A_{\mathrm{tot}}} \tag{D5}$$

with $A_{\mathrm{tot},j}$ the total surface area per unit volume of particles in the size bin $j$ and $A_{\mathrm{tot}}$ the total surface area over all size bins. In 16 freezing experiments in the AIDA cloud chamber the ice crystal number concentration, formed by active IN, $N_{\mathrm{i}}$, was measured as a function of $T$. The total particle surface area, $A_{\mathrm{tot}}$, was estimated before each experiment and multiplied with a pressure dilution factor. The evaluation of the results yields the following fit formula for the ice active surface site density of natural dust:

$$n_{\mathrm{s,IN}}(T)[\mathrm{m}^{-2}] \quad = \quad b \cdot \exp(-A \cdot a \cdot (T - 273.15 \text{ K}) + B) , \tag{D6}$$

with the fit parameters $A = -0.517$ and $B = 8.934$ and the unit correction factors $a = \mathrm{K}^{-1}$ and $b = \mathrm{m}^{-2}$. Due to the temperature range of the freezing experiments, the parameterization is limited to the temperature range from 261.15 K to 237.15 K.





Eq. D6 was used here to fit the dataset for the different mineral dust types. The fits were done in two different ways: By using Eq. D3 and Eq. D6 to fit the measured $FF$ or by using Eq. D3 to convert the $FF$ measurements to surface site densities and fit $n_{\mathrm{s,IN}}(T)$ directly following Eq. D6. The results are show in Table 6 and in Fig. 11. The scheme is labeled "$n_{\mathrm{s,IN}}$" in Fig. 5.

**Table 6.** Estimated fit parameters for the deterministic $n_{\mathrm{s,IN}}$ approach for different mineral dust types. The values are rounded to two digits after the decimal point.

| Approach | Fit parameter | Kao | Ill | Mont | Micro | ATD |
|---|---|---|---|---|---|---|
| $n_{\mathrm{s,IN}}$ direct | $A$ | 0.39 | 0.52 | - | 0.17 | 0.48 |
| (complete dataset) | $B$ | 15.62 | 11.05 | - | 25.77 | 12.23 |
| $n_{\mathrm{s,IN}}$ direct | $A$ | 0.63 | 2.35 | 1.02 | 0.24 | 1.15 |
| (only $FF$ between 0.1 and 0.9) | $B$ | 6.27 | -56.93 | -6.4 | 24.41 | -12.48 |
| $n_{\mathrm{s,IN}}$ direct | $A$ | 0.84 | 2.72 | 1.42 | 0.32 | 0.85 |
| (only $FF$ between 0.2 and 0.8) | $B$ | -1 | -70.28 | -21.24 | 21.91 | -1.13 |
| $n_{\mathrm{s,IN}}$ based on $FF$ | $A$ | 0.92 | 1.1 | 0.91 | 0.73 | 0.37 |
| (complete dataset) | $B$ | -3.77 | -10.35 | -2.71 | 9.63 | 16.04 |
| $n_{\mathrm{s,IN}}$ based on $FF$ | $A$ | 0.77 | 1.04 | 0.9 | 1.41 | 0.27 |
| (only $FF$ between 0.1 and 0.9) | $B$ | 1.12 | -8.3 | -2.11 | 8.94 | 19.2 |
| $n_{\mathrm{s,IN}}$ based on $FF$ | $A$ | 0.62 | 0.93 | 0.88 | 1.41 | 0.25 |
| (only $FF$ between 0.2 and 0.8) | $B$ | 5.97 | -4.55 | -1.5 | 8.94 | 19.84 |

Table 6 shows that the results for the fit parameters are very different depending on whether the $FF$ is fitted or the active site density $n_{\mathrm{s,IN}}$ directly. That is due to the characteristics of the freezing curve. The very small $FF$ at warm temperatures and limited $FF$ (to 1) at low temperatures leads to a flattening of the $n_{\mathrm{s,IN}}$ curve. Calculating $n_{\mathrm{s,IN}}$ at low temperatures from $FF$ close to 1 gives the number of active sites, which was needed to freeze all droplets. However it could be that more actives sites were present than needed to freeze all droplets. Therefore the tail (low and high $FF$) of the $FF$ dataset is often left out of the

fitting. Here we investigate how the fit results and the freezing curves change depending on the share of the dataset accounted for fitting. We use the complete dataset as a first step and then omit $FF$ data higher than 0.9 and lower than 0.1 and 0.8 and 0.2, respectively. Figure 12 shows this exemplary using the dataset of kaolinite particles. It can be seen in Fig. 12b that the surface site density $n_{\mathrm{s,IN}}$ is quite different depending on how it is estimated. The variation in $n_{\mathrm{s,IN}}$ depending on the share of the $FF$ dataset is larger when $n_{\mathrm{s,IN}}$ is estimated directly by fitting to calculated $n_{\mathrm{s,IN}}$ from $FF$ measurements. The largest

deviation from all other fits originates when $n_{\mathrm{s,IN}}$ is estimated directly taking all data into account. Cutting away the tail of the $FF$ measurements leads to a very similar result when $n_{\mathrm{s,IN}}$ is estimated directly (black solid line) compared to the indirect estimate of $n_{\mathrm{s,IN}}$ using the complete $FF$ data (red dashed line). The implication of the different estimations for $n_{\mathrm{s,IN}}$ is shown in Fig. 12c for an example dataset of kaolinite ($d$=400 nm, $t$=10 s). The freezing curves from the indirect $n_{\mathrm{s,IN}}$ fit are not so different from each other and capture the measurements quite well. When cutting away the tail of the $FF$ data ($FF > 0.2$ and

$FF < 0.8$) also the freezing curve based on the direct estimated $n_{\mathrm{s,IN}}$ captures the data well and falls on the freezing curve of the indirect $n_{\mathrm{s,IN}}$ using the same share of $FF$ data. It seems necessary to cut the tails away from the $FF$ data when $n_{\mathrm{s,IN}}$





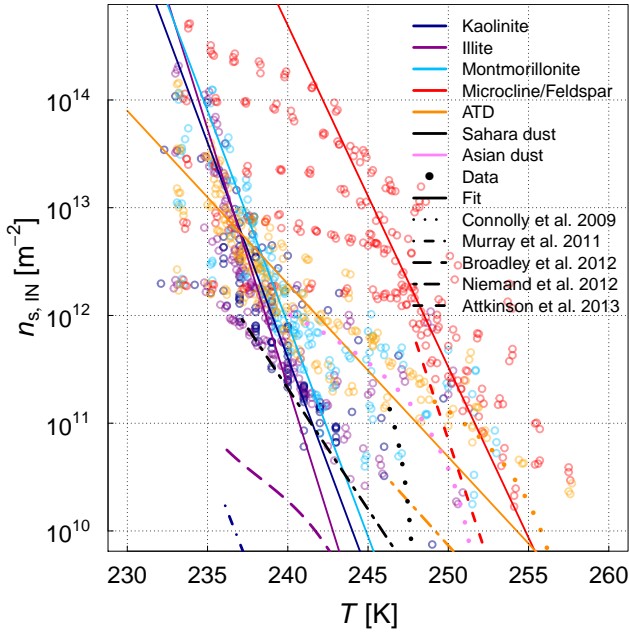

**Figure 11.** Surface site density $n_{s,IN}$ of different mineral dust types. The points are calculated surface site densities from $FF$ measurements, the lines are the corresponding fits. Some estimates of $n_{s,IN}$ found in literature are added.

is fitted directly. When estimating $n_{s,IN}$ indirectly by using $FF$ it seems that there is no need for cutting the tail. Not cutting the tail increases the amount of data available for the fitting and might therefore be preferable. However, very low/high $FF$ are most susceptible to experimental uncertainties, which could be a legitimation of cutting the tail away from the dataset. Because the results are different depending on the methodology this sensitivity should be taken into account when comparing different

5  $n_{s,IN}$ from literature. Maybe an uniform standard on how to derive $n_{s,IN}$ could help.





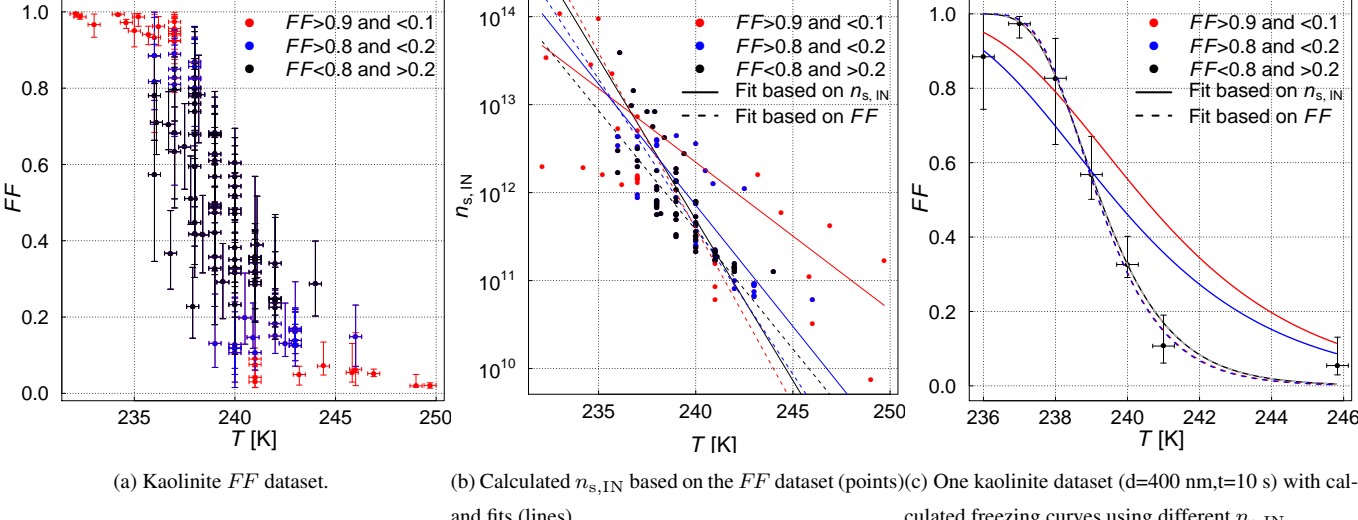

(a) Kaolinite $FF$ dataset.  (b) Calculated $n_{s,IN}$ based on the $FF$ dataset (points) (c) One kaolinite dataset ($d$=400 nm, $t$=10 s) with cal-
and fits (lines).  culated freezing curves using different $n_{s,IN}$.

**Figure 12.** Sensitivity study of different methods to estimate the surface site density $n_{s,IN}$ of kaolinite and its implications.

*Acknowledgements.* The authors would like to thank Corinna Hoose and Paul Connolly for interesting discussions and their valuable input.





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
