# Peer review of "Classical nucleation theory of immersion freezing: Sensitivity of contact angle schemes to thermodynamic and kinetic parameters"

_Atmospheric Chemistry and Physics, 2015_

## Referee Comment (RC1) · Anonymous Referee #1 · 12 Feb 2016

General comment:

In order to find an appropriate (CNT-based) parameterization for immersion freezing (induced by mineral dusts) in GCMs, the authors tested various combinations of different descriptions for interfacial free energy $\sigma_w$, activation energy $\Delta g^\#$ as well as different possibilities to include contact angle (single contact angle, contact angle distribution and temperature dependent contact angle). To do so, the different schemes are fitted against laboratory heterogeneous freezing data for different dusts (spanning several temperature, particle surface and time ranges).

I have to admit that after first reading I have been in two minds about recommending this paper for publication in ACP. On the one hand the authors e.g. vividly demon-

strate that contact angle values gained for one substance largely depend on the values/parametrizations used for $\sigma_w$ and $\Delta g^\#$ so that contact angle values obtained in different studies for a given substance do not necessarily agree. This is important if these contact angle values are used in GCMs but connected with other $\sigma_w$ and $\Delta g^\#$ values/parameterizations. On the other hand e.g. I doubt the physics behind some of the presented parameterization schemes (see specific comments). However, due to the importance for the heterogeneous ice nucleation community I would recommend this paper for publication after the following comments have been addressed.

Specific comments:

Page 1, line 10-11: I do not understand the meaning of the sentence starting with "We show that additional...". Please clarify. What is "$J_{\mathrm{imm}}$"?

Page 4, line 12-13: I agree that in case of heterogeneous ice nucleation the freezing curve is shifted to higher T compared to the homogeneous case. But does the curve necessarily has to be less steep? Looking on Fig. A3 in the paper of Hoose and Möhler (2012) it can be seen that the heterogeneous nucleation rate is steeper for higher temperatures (depending on the parameterizations used in CNT). So what causes the heterogeneous freezing curve to be less step compared to the homogeneous case?

Chapter 2.1.3: I have a problem with the interpretation of the temperature dependent contact angle scheme. In a physical sense the contact angle is "determined by the condition of mechanical equilibrium, i.e., there must be no net force component along the solid surface" (Pruppacher and Klett, 1997, P. 136). Due to the decrease of the interfacial free energies with temperature does the contact angle then should decrease with decreasing temperature for a given particle (let's assume homogeneous surface conditions)? Here, the contact angle increases with decreasing T. I would interpret this behavior in that way that particles with larger contact angles (higher energy barrier) can be activated with decreasing temperature. Is this in agreement to your description?

[Figure]

Chapter 3.1: I have some trouble with those schemes fitting $\Delta g^{\#}$. In lines 27 to 30 you mention that an aerosol type specific $\Delta g^{\#}$ value is physically questionable. On the other hand this statement is reversed on the next page saying that the particle itself might influence the diffusion of water molecules to the ice cluster. I agree that the attachment of water molecules to the ice cluster is influenced due to the presence of the ice nucleating particle i.e., the INP "blocks" water molecules since the ice cluster is just a cap and not spherical (in terms of CNT) as for the homogeneous case. But I think that $n_{\mathrm{s}}$ in CNT takes care of this. Are there any mechanisms which could confirm your hypothesis of aerosol-type specific $\Delta g^{\#}$? Is there a specific term in CNT which would take care of this? Or does $\Delta g^{\#}$ represent here just a fitting parameter without physical meaning?

On page 8, line 19-20 you mention that a single contact angle is not able to represent the experimental results for mineral dust. This has also been shown in e.g. Lüönd et al. (2010), Welti et al. (2012), etc. But is this a general finding? What about other substances like biological particles?

Page 9, line 18-20: I do not understand why this procedure has been performed. Some further explanations here would be great.

Page 10, line 26-27: The curve in Fig. 4a does not approach -100%...

Page 10, line 30: It looks like that the compensation in Fig. 4b is not completely linear.

Chapter 6 and Appendix C (especially Fig. 9), concerning particle size dependence: Do you know the reason for the change of the contact angle distribution with particle size for the kaolinite sample? In general, does the ice nucleation ability of a given substance has to scale with particle size? How pure is the used kaolinite sample as well as the other samples and is it possible that the chemical composition of the samples change with size and therefore the ice nucleating ability? Is there any bias in the measured frozen fractions due to multiple charged and therefore larger particles? A recent paper by Hartmann et al. (2016) shows that due to commonly used particle generation

methods multiple charged particles (they also used a FLUKA kaolinite sample in their study) can be present which bias the determined frozen fraction.

Conclusions: I am wondering why the alpha-pdf scheme is worse compared to the other two schemes when trying to represent the measured frozen fractions as a function of temperature for the various dusts and dust sizes. But in contrast, the time dependence of freezing can only be reasonably represented by the alpha-pdf scheme. In the latter case this was shown for a given dust (kaolinite) and one size only. The questions arises again: Is there an influence of size dependent composition or multiple charged particles (see comment above) which bias the fitting as a function of T and size?

In general, the conclusion is too vague. From my point of view, this study is a good contribution in order to find appropriate parameterizations based on CNT for GCMs. But I would suggest to clearly state the limits of parameterizing the immersion freezing behavior of mineral dusts due to e.g., limited amount of data available, bias due to possible multiple charges/size-dependent particle composition issues, etc. and that further experimental studies as a function of temperature and time are needed (only one sentence, the last one on page 15, is not sufficient).

Appendix: In general, there is no warm or cold temperature. The temperature can only be high or low. In some cases, plots/tables are shown based on $f$, others are based on contact angle $\alpha$. It is difficult to directly inter-compare the results. Could you please refer to only one parameter or both $f$ and $\alpha$?

Technical notes:

Page 9, line 4: the schemes are mixed up here: CNT #7 is the $\alpha(T)$ scheme and CNT #5 and #6 are the $\alpha$-pdf schemes.

Page 12, line 18: Should it read "..., which was found to not represent..."?

Page 15, line 6: 'have been compiled' instead of 'are compiled'?

In the upper left plots of Fig. 7 and 8 the highest temperature value (i.e., 260K) is truncated.

Figure 8, lower right panel: The mean freezing temperature is about 237K. Does this "agree" with the high mean contact angle of 0.3?

Caption of Fig. 8: There is a 'decreasing' missing in the sentence starting with: "contact angle gets larger with *decreasing* T because..."

On page 33, line 5-6. There is the verb missing in the sentence "... $FF$ is fitted or the active site density directly."

Fig. 11: The color coding is less than ideal. It is difficult to see which of the lines correspond to which study.

References:

Hartmann, S., Wex, H., Clauss, T., Augustin-Bauditz, S., Niedermeier, D., Rösch, M., and Stratmann, F. (2016), Immersion Freezing of Kaolinite: Scaling with Particle Surface Area, J. Atmos. Sci., 73(1), 263-278.

Hoose, C. and Möhler, O. (2012), Heterogeneous ice nucleation on atmospheric aerosols: a review of results from laboratory experiments, Atmos. Chem. Phys., 12, 9817-9854.

Lüönd, F., Stetzer, O., Welti, A., and Lohmann, U. (2010), Experimental study on the ice nucleation ability of size selected kaolinite particles in the immersion mode, J. Geophys. Res. – Atmos., 115(D14).

Pruppacher, H. R. and Klett, J. D. (1997), Microphysics of Clouds and Precipitation, 2. Edn., Kluwer Academic Publishers, Dordrecht.

Welti, A., Lüönd, F., Kanji, Z. A., Stetzer, O., and Lohmann, U. (2012), Time dependence of immersion freezing: an experimental study on size selected kaolinite parti-
cles, Atmos. Chem. Phys., 12(20), 9893-9907.

---

## Short Comment (SC1) · 7 Mar 2016

Peter A. Alpert

**General Comments**

The manuscript by Ickes et al. details the ability of different immersion freezing parameterizations based on classical nucleation theory to reproduce laboratory data. The authors target the treatment of the contact angle,  $\alpha$ , the interfacial tension between an ice germ and water,  $\sigma_{iw}$ , and the activation energy barrier,  $\Delta g^{\#}$ . Performance of freezing parameterizations are evaluated in terms of their accuracy in reproducing measured frozen droplet fractions (criteria 1), their consistency with trends in freezing as a function of temperature, T, particle size, and time, t, (criteria 2) and finally if their fit parameters and fitting functions lead to reasonable representations (criteria 3). Comparison of formulations 1), 5) and 7) use the same thermodynamic parameterizations for  $\sigma_{iw}$  (Reinhardt and Doye, 2013) and  $\Delta g^{\#}$  (Zobrist et al., 2007), but  $\alpha$  is represented by different schemes namely the single- $\alpha$ in 1), the  $\alpha$ -PDF in 5) and the  $\alpha(T)$  in 7).

The manuscript in question is well written and comprehensive. The overall conclusions are justified. However, a discussion of previous literature that pertains to the performance of freezing parameterizations is lacking and alternative explanations of freezing parameterization should be explored for completeness. The authors cite Marcolli et al. (2007) and Lüönd et al. (2010) on p. 4, l. 17, however, significant advancement in comparing contact angle schemes should also be discussed and attributed to Zobrist et al. (2007), Alpert et al. (2011), Knopf and Forrester (2011), Rigg et al. (2013) and Wheeler et al. (2014). I feel that a concise discussion of the major findings in these previous studies and how the current study in question advances these findings should be presented.

- In Zobrist et al. (2007), a single- $\alpha$  scheme was evaluated for performance, but it could not reproduce the freezing of droplets due to 1-nonadecanol, an organic monolayer coating. Allowing  $\alpha$  to be a function of T, i.e. the  $\alpha(T)$  scheme, resulted in a good representation of their data (Zobrist et al., 2007). However, here the authors do not mention that the single- $\alpha$  or  $\alpha(T)$  scheme was considered in Zobrist et al. (2007). In our recent publication (Alpert and Knopf, 2016), we address the question of the applicability of the single- $\alpha$  scheme to an uneven mineral dust surface, if it does not apply to uniform surface on a molecular level such as a self-assembled organic monolayer coating. In other words, a single- $\alpha$  was never shown to reproduce freezing data for a highly ordered and uniform surface. It has also been argued that  $\alpha$  represents the balance of surface tensions which can change as T changes. Thus a single- $\alpha$  scheme is not expected to be physically applicable (Welti et al., 2012; Rigg et al., 2013).

- In the studies by Alpert et al. (2011), Knopf and Forrester (2011) and Rigg et al. (2013) investigating different ice nucleating particle (INP) types, it was shown that the  $\alpha(T)$  scheme can be applied and that a linear function may be used. This corroborates findings by Zobrist et al. (2007). However, the authors neglect discussion of these studies and refer only to Welti et al. (2012).
- The authors claim that parameterizations should be consistent with freezing trends of known microphysical processes (critera 3). One easily accessible microphysical process that can be discussed for evaluating freezing parameterizations is the ability to reproduce freezing point depression of aqueous solution droplets. Rigg et al. (2013) evaluated the applicability of the  $\alpha$ -PDF and  $\alpha(T)$  scheme to describe droplet freezing experiments. In their study, organic particles were immersed in pure water or aqueous ammonium sulfate solution droplets and freezing was observed as a function of T and water activity,  $a_w$ . The analysis demonstarted that one  $\alpha$ -PDF distribution could not reproduce observed freezing data, but the data could be well represented allowing  $\alpha$  to be a function of T and  $a_w$  Rigg et al. (2013). The  $\alpha$ -PDF scheme failed as a physical representation of the ice nucleating ability of the particles. If an  $\alpha$ -PDF scheme is chosen to be used to describe immersion freezing in future studies, it should be modified to account for changes in  $a_w$  (Rigg et al., 2013). I suggest to include this discussion adding more detail to the authors' third criteria, how our current knowledge of microphysics can lead to more correct ice nucleation parameterizations and not only just better performance.
- Wheeler et al. (2014) evaluated many different  $\alpha$  schemes and found that the  $\alpha$ -PDF is not the best performing. Instead, a scheme known as the "active site scheme" (AS) is the best performing. This finding should be discussed in the manuscript on p. 5, 1. 28-p. 6, 1. 2.

**Specific Comments**

When introducing the  $\alpha$  schemes, there are a few instances where the author claims certainty or implies that INP surfaces have variable ice nucleation efficiency. There is no physical evidence that an INP has variable ice active sites or surfaces with different ice nucleation efficiency. For now, any evidence is a product of circumstance to a conceptual mathematical framework (or fitting procedure with prior assumptions of the existence of active sites). On p. 5, 1. 2-3, the authors claim that the single  $\alpha$  scheme "assumes" that the surface has one  $\alpha$  values for the entire particle surface. However,

on p. 5, l. 10, the authors claim that the  $\alpha$ -PDF "accounts" for surface heterogeneity. This later statement is incorrect. The  $\alpha$ -PDF does not account for anything, but it does *assume* that the surface of particles is covered by sites that have different contact angles. In lack of in situ observations, this is not a better or more accurate assumption, but simply a different conceptual framework. The authors should state the assumptions of all schemes accurately just as they did for the single  $\alpha$  scheme.

On p. 5, l. 4-6, the authors claim that the  $\alpha(T)$  scheme does not take into account how contact angles are distributed. Then say on l. 6-7 that good IN freeze first, e.g. when performing a cooling rate experiment, which shifts the mean contact angle of the remaining droplet population. These statements are contradictory as it is written. The authors say that the first scheme does not distribute contact angles, but the contact angle distribution shifts? Again, there is no certainty that a contact angle distribution exists in the first place. Rather it is sufficient to say that the  $\alpha(T)$  scheme assumes a physical dependence of  $\alpha$  on T. To describe this scheme as a compromise is also incorrect as it is different than the single  $\alpha$  and  $\alpha$ -PDF schemes. It it is based on different assumptions and includes a physical dependence of thermodynamic parameters on  $\alpha$  which is neglected in the other two schemes. Similarly, it also does not "reflect a changing  $\alpha$ -PDF distribution" (1. 5-6).

The authors emphasize computational efficiency, cost, expense, complexity... as a way of evaluating each scheme or parameterization. Since this is presented as a sort of metric for comparison, it should be a quantifiable metric. As it is presented by the authors here, it is not quantitative. How much time does it take for a computer to calculate  $\sigma_{iw}$  derived by the different parameterizations presented here? What is the extra time it takes to randomly sample from an  $\alpha$ -PDF or calculate  $\alpha(T)$ before freezing is predicted in a GCM? Understandably, the time it takes to fit various  $\alpha$  schemes and other parameters is very different and may take hours is some cases. After finding all parameters and using them in formulations 1-7, how long does each take to predict ice nucleation in GCM's for the same aerosol population and thermodynamic conditions? If the authors choose to not consider the active site scheme and the soccer ball model, they must have some reason and quantitative evidence as to why. For example, p. 5 1. 31-p. 6 1. 32 claims that a scheme is too computationally expensive to be considered, but no quantitative measure is given. In order for this statement and all others like it to remain in the manuscript, the authors must provide quantitative evidence for this. Is it possible to add another column in Table 2 for this purpose?

**References**

- Alpert, P. A. and Knopf, D. A.: Analysis of isothermal and cooling-rate-dependent immersion freezing by a unifying stochastic ice nucleation model, Atmos. Chem. Phys., 16, 2083–2107, doi:10.5194/acp-16-2083-2016, 2016.
- Alpert, P. A., Knopf, D. A., and Aller, J. Y.: Ice nucleation from aqueous NaCl particles with and without marine diatoms, Atmos. Chem. Phys., 11, 5539–5555, doi:10.5194/acp-11-5539-2011, 2011.
- Knopf, D. A. and Forrester, S.: Freezing of water and aqueous NaCl droplets coated by organic monolayers as a function of surfactant properties and water activity, J. Phys. Chem. A, 115, 5579–5591, doi:10.1021/jp2014644, 2011.
- Lüönd, F., Stetzer, O., Welti, A., and Lohmann, U.: Experimental study on the ice nucleation ability of size-selected kaolinite particles in the immersion mode, J. Geophys. Res., 115, D14201, doi:10.1029/2009JD012959, 2010.
- Marcolli, C., Gedamke, S., Peter, T., and Zobrist, B.: Efficiency of immersion mode ice nucleation on surrogates of mineral dust, Atmos. Chem. Phys., 7, 5081–5091, doi:10.5194/acp-7-5081-2007, 2007.
- Reinhardt, A. and Doye, J. P. K.: Note: Homogeneous TIP4P/2005 ice nucleation at low supercooling, J. Chem. Phys., 139, doi:10.1063/1.4819898, 2013.
- Rigg, Y. J., Alpert, P. A., and Knopf, D. A.: Immersion freezing of water and aqueous ammonium sulphate droplets initiated by Humic Like Substances as a function of water activity, Atmos. Chem. Phys., 13, 4917– 4961, doi:10.5194/acpd-13-4917-2013, 2013.
- Welti, A., Lüönd, F., Kanji, Z. A., Stetzer, O., and Lohmann, U.: Time dependence of immersion freezing: an experimental study on size selected kaolinite particles, Atmos. Chem. and Phys., 12, 9893–9907, doi:10.5194/acp-12-9893-2012, www.atmos-chem-phys.net/12/9893/2012/, 2012.
- Wheeler, M. J., Mason, R. H., Steunenberg, K., Wagstaff, M., Chou, C., and Bertram, A. K.: Immersion freezing of supermicron mineral dust particles: Freezing results, testing different schemes for describing ice nucleation, and ice nucleation active site densities, J. Phys. Chem. A, 119, 4358–4372, doi:10.1021/jp507875q, 2014.
- Zobrist, B., Koop, T., Luo, B. P., Marcolli, C., and Peter, T.: Heterogeneous ice nucleation rate coefficient of water droplets coated by a nonadecanol monolayer, J. Phys. Chem. A, 111, 2149–2155, doi:10.1021/jp066080w, 2007.

---

## Referee Comment (RC2) · Anonymous Referee #2 · 12 Apr 2016

The authors do a thorough examination of different possible ways of fitting the classical nucleation theory to immersion freezing experiments using various formulations for parameters of the theory. They compare the outcomes using three criteria: 1) How well each CNT formulation reproduces the experimental freezing curves; 2) How good are the size and time dependences of each formulation compared with experimental data (if available)? and 3) Are the values of the fit parameters microphysically reasonable?

I believe that this is a useful paper that should eventually be published: at the moment CNT is the only theoretically based approach that can be used as a basis for parametrizing immersion freezing for global models, and this paper provides valuable information for constructing such parametrizations. At the same time it should be kept

in mind that CNT is by no means perfect.

I have really just one major comment. The authors make the following statement in the Conclusions: "Criterion 3 is difficult to evaluate coming from a macroscopic level as microphysical knowledge is missing at this point". I don't think the situation is quite that bad. There is information available in the literature that can help at least discuss whether the contact angle distributions are physically reasonable, or consistent with information gained from other studies.

First, there are molecular dynamics papers that have investigated water (both liquid and ice) especially at kaolinite surfaces (e.g. Hu and Michaelidis, Surface Science 601, 5378, 2007; ibid, 602, 960, 2008; Croteau et al., J.Phys. Chem A, 114, 8396, 2010; Solc et al., Geoderma 169, 47, 2011). Please check these and discuss how realistic the contact angle distributions derived in your study are.

Secondly, there are observations of freezing microdroplets at different hydrophopbic surfaces indicating that the contact angle does not change when freezing occurs (Jung et al., Langmuir 27, 3059, 2011; Heydari et al., J. Phys. Chem. C, 117, 21752, 2013). This information can be used in the context of Young equations. If you write down Young equations for contact angles of 1) a water cluster on a surface S, 2) an ice cluster on S (against air) and 3) an ice cluster on S, immersed in water, you can figure out what the contact angle of ice immersed in water should be if the contact angle of liquid water on the same surface is known (literature values of water contact angles can be found many minerals).

Thirdly, one can make the following question (this time disregarding the assumption of equal contact angles for water and ice): Are the results in this work (i.e. contact angles of ice immersed in water) consistent (again, in the context of Young equations) with ice contact angles derived from deposition nucleation studies? This question should be answerable with the help of the different interfacial tensions used (water, ice, ice in water) and water contact angles. I suggest that you do these two exercises with the

Young equations and discuss.

Minor:

- Contact angle values in Tables 3 and 5 don't seem consistent with the contact angle distributions in Fig. 9. Should the radian values be multiplied by pi? However, even in that case, the mu-values in Fig. 10 appear a bit strange: they are mostly below 90 degrees, although the modes of the distributions in Fig. 9 are all above 90 degrees, and the distributions appear to be skewed right. How come?

- What happened to Appendix A?

- The order of Figs. 9 and 10 should be changed

- The English of the ms should be checked.

---

## Author Comment (AC1)

**Reviews for Atmos. Chem. Phys. Discuss., doi:10.5194/acp-2015-969, 2016.**

Ickes Luisa1\*, Welti André1+, and Lohmann Ulrike1 1IAC, ETH Zurich \*now at: IMK-AAF, KIT Karlsruhe +now at: TROPOS Leipzig

Correspondence to: Luisa Ickes (luisa.ickes@env.ethz.ch)

We thank both anonymous reviewers and P. A. Alpert for their positive review and the detailed comments on the manuscript. We have revised the manuscript accordingly (see track-changes in the manuscript). Our replies to your comments are given below in blue after the specific comment.

**1 Review 1**

**5 1.1 General comment:**

In order to find an appropriate (CNT-based) parameterization for immersion freezing (induced by mineral dusts) in GCMs, the authors tested various combinations of different descriptions for interfacial free energy  $\sigma_w$ , activation energy  $\Delta g^{\#}$  as well as different possibilities to include contact angle (single contact angle, contact angle distribution and temperature dependent contact angle). To do so, the different schemes are fitted against laboratory heterogeneous freezing data for different dusts (spanning several temperature, particle surface and time ranges).

I have to admit that after first reading I have been in two minds about recommending this paper for publication in ACP. On the one hand the authors e.g. vividly demonstrate that contact angle values gained for one substance largely depend on the values/parametrizations used for  $\sigma_w$  and  $\Delta g^{\#}$  so that contact angle values obtained in different studies for a given substance do not necessarily agree. This is important if these contact angle values are used in GCMs but connected with other  $\sigma_w$  and  $\Delta g^{\#}$

15 values/parameterizations. On the other hand e.g. I doubt the physics behind some of the presented parameterization schemes (see specific comments). However, due to the importance for the heterogeneous ice nucleation community I would recommend this paper for publication after the following comments have been addressed.

We attempt to clarify the physics behind some of the presented parameterization schemes below. However, the microphysical details how freezing of supercooled water is triggered heterogeneously are not very well understood so that some approaches

20

10

are based on speculative assumptions. It is not the aim of this article to judge how reasonable the assumptions are, but to show different strategies to handle heterogeneous freezing with CNT based parametrisations and to investigate how well these parameterizations describe a selection of laboratory data.

**1.2** Specific comments:**

Page 1, line 10-11: I do not understand the meaning of the sentence starting with "We show that additional...". Please clarify. What is " $J_{imm}$ "?

We changed that in the manuscript. We also added an explanation for  $J_{imm}$ .

- 5 Page 4, line 12-13: I agree that in case of heterogeneous ice nucleation the freezing curve is shifted to higher T compared to the homogeneous case. But does the curve necessarily has to be less steep? Looking on Fig. A3 in the paper of Hoose and Möhler (2012) it can be seen that the heterogeneous nucleation rate is steeper for higher temperatures (depending on the parameterizations used in CNT). So what causes the heterogeneous freezing curve to be less step compared to the homogeneous case?
- 10 The temperature dependence of the nucleation rate Jimm (or Jhom) is controlled by the temperature dependence of the Gibbs free energy barrier ΔG and the activation energy barrier Δg#, which have opposite temperature dependences. In the case of immersion freezing ΔG is reduced and therefore the contribution of Δg# is more important, which leads to a flattening of the heterogeneous nucleation curve compared to homogeneous nucleation. Looking at measurements of the frozen fraction as a function of T (homogeneous freezing compared to immersion freezing) one can see that indeed the curves for immersion 15 freezing are less steep compared to homogeneous freezing.

Evaluating the steepness of different curves relative to each other is also a matter of perspective. Looking at the figure below it can be seen that when comparing the curves within the red box, the steepness increases with decreasing f (hom. freezing vs. het. freezing). Looking at the curves within the same temperature range (blue box), however, the steepness of the curve decreases in case of heterogeneous freezing. The second case is the situation we were referring to. We added a better explanation in the manuscript.

20

Chapter 2.1.3: I have a problem with the interpretation of the temperature dependent contact angle scheme. In a physical sense the contact angle is "determined by the condition of mechanical equilibrium, i.e., there must be no net force component along the solid surface" (Pruppacher and Klett, 1997, P. 136). Due to the decrease of the interfacial free energies with temperature does the contact angle then should decrease with decreasing temperature for a given particle (let's assume homogeneous

5 surface conditions)? Here, the contact angle increases with decreasing T. I would interpret this behavior in that way that particles with larger contact angles (higher energy barrier) can be activated with decreasing temperature. Is this in agreement to your description?

Yes, this is in agreement. The  $\alpha(T)$  scheme accounts for the change of the contact angles being activated at different T resulting in a shift of the contact angle distribution towards larger average contact angle with supercooling. The lower the temperature, the higher the chance that particles with larger contact angles can be activated. We clarified that in the text.

10

15

Chapter 3.1: I have some trouble with those schemes fitting  $\Delta g^{\#}$ . In lines 27 to 30 you mention that an aerosol type specific  $\Delta g^{\#}$  value is physically questionable. On the other hand this statement is reversed on the next page saying that the particle itself might influence the diffusion of water molecules to the ice cluster. I agree that the attachment of water molecules to the ice cluster is influenced due to the presence of the ice nucleating particle i.e., the INP "blocks" water molecules since the ice cluster is just a cap and not spherical (in terms of CNT) as for the homogeneous case. But I think that  $n_s$  in CNT takes care of

this. Are there any mechanisms which could confirm your hypothesis of aerosol-type specific  $\Delta g^{\#}$ ? Is there a specific term in CNT which would take care of this? Or does  $\Delta g^{\#}$  represent here just a fitting parameter without physical meaning?

 $n_{\rm s}$  is the number of water molecules in contact with the unit area of an ice germ and is mostly estimated based on the molecular density of water. The activation energy  $\Delta g^{\#}$  describes the diffusion of a water molecule across the water-ice boundary. Most

- 20 authors assume that this diffusion process for immersion freezing is equivalent to the one for homogeneous freezing. In case of heterogeneous freezing the diffusion could be disturbed by the presence of IN in the water (surface charges, polarizability of the particle surface, etc.). Our hypothesis is that the activation energy  $\Delta g^{\#}$  is different within a pure supercooled water droplet compared to a supercooled water droplet containing an insoluble aerosol particle (IN), because the IN (eventually containing surface charges) effects the hydrogen bond network and due to that could influence the diffusion process. One example study
- 25 looking at the influence of surface charge on ice nucleation is the study of Edwards and Evans (1962). We added this reference.

On page 8, line 19-20 you mention that a single contact angle is not able to represent the experimental results for mineral dust. This has also been shown in e.g. Lüönd et al. (2010), Welti et al. (2012), etc. But is this a general finding? What about other substances like biological particles?

30 We only investigated the behavior of a single contact angle to describe heterogeneous freezing of mineral dust. We did not look at biological particles, therefore we can not make a general statement about the ability of a single contact angle to parameterize their freezing. Also ice nucleation by AgI seems to be captured by a single alpha. Possibly a single alpha scheme can be used for highly efficient IN triggering ice formation at low supercooling. Page 9, line 18-20: I do not understand why this procedure has been performed. Some further explanations here would be great.

This exercise was done to exemplary show the consequence of using fit parameters together with a CNT formulation, which was not the one used to derive the fit parameters. We included an additional explanation to make this clearer.

5 Page 10, line 26-27: The curve in Fig. 4a does not approach -100%...

We only show the part of the curve where changes are large. The plot with an increased range is added below.

Page 10, line 30: It looks like that the compensation in Fig. 4b is not completely linear.

Thanks for recognizing this, we did not see that at the first glance. While the deviation of  $\mu$  with variation of  $\Delta g^{\#}$  is linear, the deviation of  $\sigma$  is linear for a small variation of  $\Delta g^{\#}$  (until 30% approximately), but is nonlinear for larger variations of  $\Delta g^{\#}$ . We adapted the text accordingly.

Chapter 6 and Appendix C (especially Fig. 9), concerning particle size dependence: Do you know the reason for the change of the contact angle distribution with particle size for the kaolinite sample? In general, does the ice nucleation ability of a given substance has to scale with particle size? How pure is the used kaolinite sample as well as the other samples and is it possible that the chemical composition of the samples change with size and therefore the ice nucleating ability? Is there any bias in the

15 measured frozen fractions due to multiple charged and therefore larger particles? A recent paper by Hartmann et al. (2016)

shows that due to commonly used particle generation methods multiple charged particles (they also used a FLUKA kaolinite sample in their study) can be present which bias the determined frozen fraction.

No, we do not know for sure the reason for the change of the contact angle distribution. In general we think that it is not unlikely that the ice nucleation ability of a given substance changes with size- that can also be seen in the active sites scheme,

5 where an increase in size of the IN increases the chance that this IN contains an active site to trigger freezing at a specific temperature. From Fig. 9 one can thus conclude that an increase in size of the IN leads to a wider spread of possible contact angles (wider alpha-pdf distribution) and to a shift to smaller contact angles with size (because the chance to have a small contact angle increases).

For solid IN, the larger the surface area the higher the freezing temperature. This is generally the case but the change in 10 freezing temperature with size depends logarithmic on the surface area and therefore approaches a constant temperature.

The used kaolinite sample was relatively pure, the contamination was less than 10%. Such a low contamination does not show any effect on the measurement and is negligible. We can not answer the question if the contamination level significantly changed with size.

- The size selection was done with caution to prevent multiple charges. An elaborate size selection setup was used: a cascade 15 of impactors and cyclones to reduce the amount of large particles after aerosol generation and neutralize the particles by impactions of ions followed by size selection with a Differential Mass Analyzer (DMA). However, multiple charges exist in the smaller size range (mainly 100 and 200 nm). The larger the particle the less they are effected by multiple charges. At a later state a CPMA was additionally included in the setup to make sure that no multiple charges exist. The mentioned updated setup was used for the microcline measurements presented in this paper.
- 20 The multiple charges were estimated for the kalonite sample (Lüönd et al. 2012?). Correcting the size accordingly and refitting the dataset shows that the uncertainty due to multiple charges is not significant. The new curves lie within the error-range of the data points. Only for the 100 and 200 nm dataset there is some deviation from the original curve.

Conclusions: I am wondering why the alpha-pdf scheme is worse compared to the other two schemes when trying to represent the measured frozen fractions as a function of temperature for the various dusts and dust sizes. But in contrast, the time

25 dependence of freezing can only be reasonably represented by the alpha-pdf scheme. In the latter case this was shown for a given dust (kaolinite) and one size only. The questions arises again: Is there an influence of size dependent composition or multiple charged particles (see comment above) which bias the fitting as a function of T and size?

Since we only have size and time dependent measurements in case of kaolinite, it is difficult to say that the alpha-pdf scheme is in general worse than the two others. However, the reason that the formulas give better fit results in the other two cases could be only mathematical. The effect of multiple charges is negligible for the results (see answer above).

30

In general, the conclusion is too vague. From my point of view, this study is a good contribution in order to find appropriate parameterizations based on CNT for GCMs. But I would suggest to clearly state the limits of parameterizing the immersion freezing behavior of mineral dusts due to e.g., limited amount of data available, bias due to possible multiple charges/sizedependent particle composition issues, etc. and that further experimental studies as a function of temperature and time are needed (only one sentence, the last one on page 15, is not sufficient).

Thanks for pointing this out. We emphasized the limits/lack of experimental studies more in the conclusion.

Appendix: In general, there is no warm or cold temperature. The temperature can only be high or low. In some cases,
plots/tables are shown based on *f*, others are based on contact angle α. It is difficult to directly inter-compare the results. Could you please refer to only one parameter or both *f* and α?

Indeed, thanks, this is corrected. The reason for showing sometimes f, sometimes  $\alpha$  on the plots/tables is that the fit parameter itself is different in the different cases. It is possible to convert f to  $\alpha$  using Eq. 3. We added the converted/approximated values of f to make a comparison easier.

**10 **1.3 Technical notes:**

Page 9, line 4: the schemes are mixed up here: CNT #7 is the  $\alpha$ (T) scheme and CNT #5 and #6 are the  $\alpha$ -pdf schemes.

This is corrected.

Page 12, line 18: Should it read "..., which was found to not represent..."?

Yes, thanks.

15 Page 15, line 6: "have been compiled" instead of "are compiled"?

**Done.**

In the upper left plots of Fig. 7 and 8 the highest temperature value (i.e., 260 K) is truncated.

**Thanks for pointing that out, that is fine now.**

Figure 8, lower right panel: The mean freezing temperature is about 237 K. Does this "agree" with the high mean contact 20 angle of 0.3?

There was a mistake in the plotting script for Fig. 7 and 8, thanks for noticing. We corrected it. Additionally we changed the values for the fixed variables (like  $\mu$ ) to values, which are closer to the resulting fit parameters from Table 3. The mean freezing temperature for the  $\alpha$ -pdf scheme when using a mean contact angle of 0.5 (Fig. 8 lower panel) is approximately 239 K. That goes along with Fig. 5 (fits and measurement points for kaolinite).

25 Caption of Fig. 8: There is a "decreasing" missing in the sentence starting with: "contact angle gets larger with *decreasing* T because...".

**Yes, thanks.**

On page 33, line 5-6. There is the verb missing in the sentence "... FF is fitted or the active site density directly."

This is corrected, thanks.

Fig. 11: The color coding is less than ideal. It is difficult to see which of the lines correspond to which study.

The colors are changed now.

**1.4 References:**

10

5 Hartmann, S., Wex, H., Clauss, T., Augustin-Bauditz, S., Niedermeier, D., Rösch, M., and Stratmann, F. (2016), Immersion Freezing of Kaolinite: Scaling with Particle Surface Area, J. Atmos. Sci., 73(1), 263-278.

Hoose, C. and Möhler, O. (2012), Heterogeneous ice nucleation on atmospheric aerosols: a review of results from laboratory experiments, Atmos. Chem. Phys., 12, 9817-9854.

Lüönd, F., Stetzer, O., Welti, A., and Lohmann, U. (2010), Experimental study on the ice nucleation ability of size selected kaolinite particles in the immersion mode, J. Geophys. Res. – Atmos., 115(D14).

Pruppacher, H. R. and Klett, J. D. (1997), Microphysics of Clouds and Precipitation, 2. Edn., Kluwer Academic Publishers, Dordrecht.

Welti, A., Lüönd, F., Kanji, Z. A., Stetzer, O., and Lohmann, U. (2012), Time dependence of immersion freezing: an experimental study on size selected kaolinite particles, Atmos. Chem. Phys., 12(20), 9893-9907.

**References**

Edwards, G. R. and Evans, L. F.: Effect of surface charge on ice nucleation by silver iodide, Trans. Faraday Soc., 58, 1649–1655, 1962.

---

## Author Comment (AC2)

**Reviews for Atmos. Chem. Phys. Discuss., doi:10.5194/acp-2015-969, 2016.**

Ickes Luisa1\*, Welti André1+, and Lohmann Ulrike1

1IAC, ETH Zurich \*now at: IMK-AAF, KIT Karlsruhe +now at: TROPOS Leipzig *Correspondence to:* Luisa Ickes (luisa.ickes@env.ethz.ch)

We thank both anonymous reviewers and P. A. Alpert for their positive review and the detailed comments on the manuscript. We have revised the manuscript accordingly (see track-changes in the manuscript). Our replies to your comments are given below in blue after the specific comment.

**1 Interactive comment 1 by Peter A. Alpert**

**5 1.1 General Comments**

The manuscript by Ickes et al. details the ability of different immersion freezing parameterizations based on classical nucleation theory to reproduce laboratory data. The authors target the treatment of the contact angle,  $\alpha$ , the interfacial tension between an ice germ and water,  $\sigma_{iw}$ , and the activation energy barrier,  $\Delta g^{\#}$ . Performance of freezing parameterizations are evaluated in terms of their accuracy in reproducing measured frozen droplet fractions (criteria 1), their consistency with trends in freezing

- 10 as a function of temperature, T, particle size, and time, t, (criteria 2) and finally if their fit parameters and fitting functions lead to reasonable representations (criteria 3). Comparison of formulations 1), 5) and 7) use the same thermodynamic parameterizations for  $\sigma_{iw}$  (Reinhardt and Doye, 2013) and  $\Delta g^{\#}$  (Zobrist et al., 2007), but  $\alpha$  is represented by different schemes namely the single- $\alpha$  in 1), the  $\alpha$ -PDF in 5) and the  $\alpha$ (T) in 7).
- The manuscript in question is well written and comprehensive. The overall conclusions are justified. However, a discussion of previous literature that pertains to the performance of freezing parameterizations is lacking and alternative explanations of freezing parameterization should be explored for completeness. The authors cite Marcolli et al. (2007) and Lüönd et al. (2010) on p. 4, l. 17, however, significant advancement in comparing contact angle schemes should also be discussed and attributed to Zobrist et al. (2007), Alpert et al. (2011), Knopf and Forrester (2011), Rigg et al. (2013) and Wheeler et al. (2014). I feel that a concise discussion of the major findings in these previous studies and how the current study in question advances these findings should be presented.
- 20 findings should be presented.

Thanks for that comment. We added a section on previous literature to point out how this study conplements the earlier findings. The aim of the current manuscript is to evaluate how suitable four different CNT based schemes are for implementation in a GCM. A broad review on ice nucleation parameterizations is not within the scope of the current work.

- In Zobrist et al. (2007), a single- $\alpha$  scheme was evaluated for performance, but it could not reproduce the freezing of droplets due to 1-nonadecanol, an organic monolayer coating. Allowing  $\alpha$  to be a function of T, i.e. the  $\alpha(T)$  scheme, resulted in a good representation of their data (Zobrist et al., 2007). However, here the authors do not mention that the single- $\alpha$  or  $\alpha$ (T) scheme was considered in Zobrist et al. (2007). In our recent publication (Alpert and Knopf, 2016), we address the question of the applicability of the single- $\alpha$  scheme to an uneven mineral dust surface, if it does not apply to uniform surface on a molecular level such as a self-assembled organic monolayer coating. In other words, a single- $\alpha$ was never shown to reproduce freezing data for a highly ordered and uniform surface. It has also been argued that  $\alpha$ represents the balance of surface tensions which can change as T changes. Thus a single- $\alpha$  scheme is not expected to be physically applicable (Welti et al., 2012; Rigg et al., 2013).
- Differently to Zobrist et al. 2007, the schemes are evaluated to reproduce ice nucleation measurements on solid surfaces 10 of mineral dust particles. As stated in Zobrist et al. 2007: "The experimentally determined heterogeneous ice nucleation rate coefficient (of a nondecanol monolayer) shows a much weaker temperature dependence than homogeneous ice nucleation and heterogeneous freezing in the presence of a solid ice nucleus such as Al2O3". Early experiments on AgI and MD simulations e.g. Cabriolu and Li (2015) support the applicability of single-alpha in certain cases. We added the references mentioned above and some further sentences about the single- $\alpha$  scheme in the manuscript.
  - In the studies by Alpert et al. (2011), Knopf and Forrester (2011) and Rigg et al. (2013) investigating different ice nucleating particle (INP) types, it was shown that the  $\alpha(T)$  scheme can be applied and that a linear function may be used. This corroborates findings by Zobrist et al. (2007). However, the authors neglect discussion of these studies and refer only to Welti et al. (2012).
- 20 The change in contact angle with  $RH_i$  and temperature can be interpreted either as a result of the temperature dependence of the interfacial tensions ( $\sigma_{is}$  and  $\sigma_{iw}$ , where the index i stands for ice, w for water and s for the aerosol surface) or as the apparent contact angle of an ensemble with a diversity of contact angles from particle to particle. In contrast to the mentioned references we follow the second interpretation as was done in Welti et al. (2012). Therefore we only referenced this paper. We added some further explanation to the manuscript and added the references.
- 25 - The authors claim that parameterizations should be consistent with freezing trends of known microphysical processes (critera 3). One easily accessible microphysical process that can be discussed for evaluating freezing parameterizations is the ability to reproduce freezing point depression of aqueous solution droplets. Rigg et al. (2013) evaluated the applicability of the  $\alpha$ -PDF and  $\alpha$ (T) scheme to describe droplet freezing experiments. In their study, organic particles were immersed in pure water or aqueous ammonium sulfate solution droplets and freezing was observed as a function of T and water activity,  $a_{\rm w}$ . The analysis demonstarted that one  $\alpha$ -PDF distribution could not reproduce observed freezing 30 data, but the data could be well represented allowing  $\alpha$  to be a function of T and  $a_w$  (Rigg et al., 2013). The  $\alpha$ -PDF scheme failed as a physical representation of the ice nucleating ability of the particles. If an  $\alpha$ -PDF scheme is chosen to be used to describe immersion freezing in future studies, it should be modified to account for changes in  $a_w$  (Rigg et al.,

5

15

2013). I suggest to include this discussion adding more detail to the authors' third criteria, how our current knowledge of microphysics can lead to more correct ice nucleation parameterizations and not only just better performance.

**Thanks, this is added in the discussion.**

- Wheeler et al. (2014) evaluated many different  $\alpha$ -schemes and found that the  $\alpha$ -PDF is not the best performing. Instead, a
- 5

30

scheme known as the "active site scheme" (AS) is the best performing. This finding should be discussed in the manuscript on p. 5, 1, 28-p. 6, 1, 2.

This is added in the section referring to other studies from literature.

**1.2 Specific comments:**

When introducing the  $\alpha$  schemes, there are a few instances where the author claims certainty or implies that INP surfaces

- 10 have variable ice nucleation efficiency. There is no physical evidence that an INP has variable ice active sites or surfaces with different ice nucleation efficiency. For now, any evidence is a product of circumstance to a conceptual mathematical framework (or fitting procedure with prior assumptions of the existence of active sites). On p. 5, 1. 2-3, the authors claim that the single- $\alpha$  scheme "assumes" that the surface has one  $\alpha$  values for the entire particle surface. However, on p. 5, 1. 10, the authors claim that the  $\alpha$ -PDF "accounts" for surface heterogeneity. This later statement is incorrect. The  $\alpha$ -PDF does not account for anything, but
- 15 it does assume that the surface of particles is covered by sites that have different contact angles. In lack of in situ observations, this is not a better or more accurate assumption, but simply a different conceptual framework. The authors should state the assumptions of all schemes accurately just as they did for the single- $\alpha$  scheme.

You are right, all asumptions should be stated. We changed this accordingly.

On p. 5, 1. 4-6, the authors claim that the α(T) scheme does not take into account how contact angles are distributed. Then
say on 1. 6-7 that good IN freeze first, e.g. when performing a cooling rate experiment, which shifts the mean contact angle of the remaining droplet population. These statements are contradictory as it is written. The authors say that the first scheme does not distribute contact angles, but the contact angle distribution shifts? Again, there is no certainty that a contact angle distribution exists in the first place. Rather it is sufficient to say that the α(T) scheme assumes a physical dependence of α on T. To describe this scheme as a compromise is also incorrect as it is different than the single-α and α-PDF schemes. It it is based on different assumptions and includes a physical dependence of thermodynamic parameters on α which is neglected in the other two schemes. Similarly, it also does not "reflect a changing α-PDF distribution" (1. 5-6).

We rephrased the explanation of the  $\alpha(T)$  scheme used and hope that the additional explanation leads to less confusion. In our case the  $\alpha(T)$  scheme is a simplified temperature dependent  $\alpha$ -PDF scheme and not a scheme based on the physical dependence of  $\alpha$  on T. Accounting for a physical dependence of  $\alpha$  on T as a result of the temperature dependence of the interfacial tensions leads to a decrease of  $\alpha$ , which is contradictory to the assumption we made here.

The authors emphasize computational efficiency, cost, expense, complexity... as a way of evaluating each scheme or parameterization. Since this is presented as a sort of metric for comparison, it should be a quantifiable metric. As it is presented by the authors here, it is not quantitative. How much time does it take for a computer to calculate  $\sigma_{iw}$  derived by the different parameterizations presented here? What is the extra time it takes to randomly sample from an  $\alpha$ -PDF or calculate  $\alpha(T)$  before freezing is predicted in a GCM? Understandably, the time it takes to fit various  $\alpha$  schemes and other parameters is very different and may take hours is some cases. After finding all parameters and using them in formulations 1-7, how long does

- 5 each take to predict ice nucleation in GCM's for the same aerosol population and thermodynamic conditions? If the authors choose to not consider the active site scheme and the soccer ball model, they must have some reason and quantitative evidence as to why. For example, p. 5 l. 31-p. 6 l. 32 claims that a scheme is too computationally expensive to be considered, but no quantitative measure is given. In order for this statement and all others like it to remain in the manuscript, the authors must provide quantitative evidence for this. Is it possible to add another column in Table 2 for this purpose?
- 10 We agree that a quantification of computational efficiency would be a nice additional information to know. However, that is not easy to derive and was therefore not mentioned explicitly. The computational time for certain freezing schemes depend on many aspects, amongst others the GCM used (treatment of aerosol particles, microphysics etc.) and can therefore not be generalised. A quantitative number in Table 2 would be meaningless.
- There are some general thoughts, which can be made to assess the computational complexity of schemes before implementing 15 them, which is the number and kind of variables needded for the scheme. In that sense an  $\alpha$ (T) scheme is not different from a single- $\alpha$  scheme (same number of variables, the same number of equation have to be solved). The  $\alpha$ -PDF scheme on the other hand is therefore more difficult as it requires an extratracer (variable, which is stored in the model longer than for one timestep) if it is implemented physically correctly. That is because the model has to memorize which contact angles from the distribution were already used in the timestep before. Otherwise the same "good" IN would be used over and over again.
- 20 Using extratracer for the contact angle of mineral dust particles would approximately lead to an increase of computational costs of 21% in the GCM ECHAM6-HAM2. The depletion of contact angles can be ignored if one assumes that the aerosol particles are replenished within one time step, so that there are always the same contact angles available (if aerosol particles are available). If the time evolution of the contact angle distribution is not taken into account, the  $\alpha$ -pdf scheme becomes computationally similarly expensive as the single- $\alpha$  scheme. However, the integral of the contact angle distribution can not
- 25 be solved analytically. Therefore, to minimize computational costs, a look-up table could be used instead of discretized finite sums. Using look-up tables is depending on the size and format of the look-up table more expensive compared to solving an equation with simple constants as in the case of the single- $\alpha$  scheme. In the case of the soccer ball model it might be extensive work to create look-up tables. The computational costs are higher for the active sites scheme. It requires a memory of used contact angles in dependence of time and therefore at least one tracer variable. In many GCMs with explicit microphysics and
- 30 aerosol dynamics, tracers are one of the major sources for computational time, which makes the code costly compared to other GCMs.

We added this discussion in the revised manuscript.

**1.3 References**

Alpert, P. A. and Knopf, D. A.: Analysis of isothermal and cooling-rate-dependent immersion freezing by an unifying stochastic ice nucleation model, Atmos. Chem. Phys., 16, 2083–2107, doi:10.5194/acp-16-2083-2016, 2016.

Alpert, P. A., Knopf, D. A., and Aller, J. Y.: Ice nucleation from aqueous NaCl particles with and without marine diatoms,
5 Atmos. Chem. Phys., 11, 5539–5555, doi:10.5194/acp-11-5539-2011, 2011.

Knopf, D. A. and Forrester, S.: Freezing of water and aqueous NaCl droplets coated by organic monolayers as a function of surfactant properties and water activity, J. Phys. Chem. A, 115, 5579–5591,doi:10.1021/jp2014644, 2011.

Lüönd, F., Stetzer, O., Welti, A., and Lohmann, U.: Experimental study on the ice nucleation ability of size-selected kaolinite particles in the immersion mode, J. Geophys. Res., 115, D14 201, doi:10.1029/2009JD012959, 2010.

10 Marcolli, C., Gedamke, S., Peter, T., and Zobrist, B.: Efficiency of immersion mode ice nucleation on surrogates of mineral dust, Atmos. Chem. Phys., 7, 5081–5091, doi:10.5194/acp-7-5081-2007, 2007.

Reinhardt, A. and Doye, J. P. K.: Note: Homogeneous TIP4P/2005 ice nucleation at low supercooling, J. Chem. Phys., 139, doi:10.1063/1.4819898, 2013.

Rigg, Y. J., Alpert, P. A., and Knopf, D. A.: Immersion freezing of water and aqueous ammonium sulphate droplets initiated
by Humic Like Substances as a function of water activity, Atmos. Chem. Phys., 13, 4917–4961, doi:10.5194/acpd-13-4917-2013, 2013.

Welti, A., Lüönd, F., Kanji, Z. A., Stetzer, O., and Lohmann, U.: Time dependence of immersion freezing: an experimental study on size selected kaolinite particles, Atmos. Chem. and Phys., 12, 9893?9907, doi:10.5194/acp-12-9893-2012, www.atmos-chem-phys.net/12/9893/2012/, 2012.

20 Wheeler, M. J., Mason, R. H., Steunenberg, K., Wagstaff, M., Chou, C., and Bertram, A. K.: Immersion freezing of supermicron mineral dust particles: Freezing results, testing different schemes for describing ice nucleation, and ice nucleation active site densities, J. Phys. Chem. A, 119, 4358–4372, doi:10.1021/jp507875q, 2014.

Zobrist, B., Koop, T., Luo, B. P., Marcolli, C., and Peter, T.: Heterogeneous ice nucleation rate coefficient of water droplets coated by a nonadecanol monolayer, J. Phys. Chem. A, 111, 2149–2155, doi:10.1021/jp066080w, 2007.

**References**

Cabriolu, R. and Li, T.: Ice nucleation on carbon surface supports the classical theory for heterogeneous nucleation, Phys. Rev. E, 91, 2015. Welti, A., Lüönd, F., Kanji, Z. A., Stetzer, O., and Lohmann, U.: Time dependence of immersion freezing, Atmos. Chem. Phys., 12, 9893–9907, 2012.

---

## Author Comment (AC3)

**Reviews for Atmos. Chem. Phys. Discuss., doi:10.5194/acp-2015-969, 2016.**

Ickes Luisa[1*], Welti André[1+], and Lohmann Ulrike[1]

[1]IAC, ETH Zurich
[*]now at: IMK-AAF, KIT Karlsruhe
[+]now at: TROPOS Leipzig

*Correspondence to:* Luisa Ickes (luisa.ickes@env.ethz.ch)

We thank both anonymous reviewers and P. A. Alpert for their positive review and the detailed comments on the manuscript. We have revised the manuscript accordingly (see track-changes in the manuscript). Our replies to your comments are given below in blue after the specific comment.

**1 Review 2**

5    The authors do a thorough examination of different possible ways of fitting the classical nucleation theory to immersion freezing experiments using various formulations for parameters of the theory. They compare the outcomes using three criteria: 1) How well each CNT formulation reproduces the experimental freezing curves; 2) How good are the size and time dependences of each formulation compared with experimental data (if available)? and 3) Are the values of the fit parameters microphysically reasonable? I believe that this is a useful paper that should eventually be published: at the moment CNT is the only theoretically
10    based approach that can be used as a basis for parametrizing immersion freezing for global models, and this paper provides valuable information for constructing such parametrizations. At the same time it should be kept in mind that CNT is by no means perfect.

I have really just one major comment. The authors make the following statement in the Conclusions: "Criterion 3 is difficult to evaluate coming from a macroscopic level as microphysical knowledge is missing at this point". I don't think the situation
15    is quite that bad. There is information available in the literature that can help at least discuss whether the contact angle distributions are physically reasonable, or consistent with information gained from other studies.
First, there are molecular dynamics papers that have investigated water (both liquid and ice) especially at kaolinite surfaces (e.g. Hu and Michaelidis, Surface Science 601, 5378, 2007; ibid, 602, 960, 2008; Croteau et al., J.Phys. Chem A, 114, 8396, 2010; Solc et al., Geoderma 169, 47, 2011). Please check these and discuss how realistic the contact angle distributions derived
20    in your study are.

We read the mentioned studies with great interest. However, only in Solč et al. 2011 a microscopic contact angle was calculated, which we could compare to our results. The derived contact angle is about $105°$ (that is equivalent to a geometric term $f$ of 0.69). However, there are many assumptions needed to calculate the microscopic contact angle. Probably more

simulations and a more detailed comparison is needed in future. We added some more information about molecular dynamics paper and recent investigations in the text.

Secondly, there are observations of freezing microdroplets at different hydrophopbic surfaces indicating that the contact angle does not change when freezing occurs (Jung et al., Langmuir 27, 3059, 2011; Heydari et al., J. Phys. Chem. C, 117, 21752, 2013). This information can be used in the context of Young equations. If you write down Young equations for contact angles of 1) a water cluster on a surface S, 2) an ice cluster on S (against air) and 3) an ice cluster on S, immersed in water, you can figure out what the contact angle of ice immersed in water should be if the contact angle of liquid water on the same surface is known (literature values of water contact angles can be found many minerals).

We are not sure if we understood the suggestion correctly. We used the Young equation for immersion freezing ($\cos(\alpha) = \frac{\sigma_{\mathrm{sw}} - \sigma_{\mathrm{si}}}{\sigma_{\mathrm{iw}}}$; index s stands for surface (aerosol), i for ice and w for water) to calculate the interfacial tension between a surface (aerosol) and ice for kaolinite (Young, 1805). We got values of approx. 58 mJ/m$^2$ at 243 K increasing by approx. 0.2 mJ/m$^2$/K with decreasing temperature. Similar results were already discussed by Welti et al. (2012). Because of that and since the different contact angles coming from different contact angle schemes yield approximately the same results, we do not see much value in adding this information or discussion to the paper.

Thirdly, one can make the following question (this time disregarding the assumption of equal contact angles for water and ice): Are the results in this work (i.e. contact angles of ice immersed in water) consistent (again, in the context of Young equations) with ice contact angles derived from deposition nucleation studies? This question should be answerable with the help of the different interfacial tensions used (water, ice, ice in water) and water contact angles. I suggest that you do these two exercises with the Young equations and discuss.

We calculated the contact angle for deposition nucleation from the different contact angles for immersion freezing using the following formula:

$$cos(\alpha_{\mathrm{dep}}) = \frac{cos(\alpha_{\mathrm{imm}})\sigma_{\mathrm{iw}} + \sigma_{\mathrm{sv}} - \sigma_{\mathrm{sw}}}{\sigma_{\mathrm{vi}}} \ .$$

For kaolinite the calculated contact angle for deposition freezing at 243 K is approx. 20°.

**1.1 Minor:**

- Contact angle values in Tables 3 and 5 don't seem consistent with the contact angle distributions in Fig. 9. Should the radian values be multiplied by $\pi$? However, even in that case, the $\mu$-values in Fig. 10 appear a bit strange: they are mostly below 90 degrees, although the modes of the distributions in Fig. 9 are all above 90 degrees, and the distributions appear to be skewed right. How come?

  Thank you for spotting this. We phrased it wrongly what $\mu$ is and the illustration in Fig. 1 was also not correct. We wrote $\mu$ is the mean contact angle, but since it is a log-normal distribution $\mu$ is the ln of the mean contact angle instead. If you exponentiate $\mu$ and then multiply it with 180° and divide it by $\pi$ you end up with the values ploted in Fig. 10. We changed the explanation for $\mu$ and adapted the illustration in Fig. 1.

The curves in Fig. 10 are fully symmetric and not skewed- maybe you got this impression because the x-axis is chosen such that the purple distribution (920 nm) is cut?

– What happened to Appendix A?

We are not sure if we understand that question. Appendix A can be found on page 26 and 27 (27 and 28 in the revised version; Fig. 7 and 8). We changed the references in the text directly refering to the figures.

– The order of Figs. 9 and 10 should be changed

Done, thanks.

– The English of the ms should be checked.

We thouroughly checked the language again and hope that the manuscript reads better now.

**References**

Welti, A., Lüönd, F., Kanji, Z. A., Stetzer, O., and Lohmann, U.: Time dependence of immersion freezing, Atmos. Chem. Phys., 12, 9893–9907, 2012.

Young, T.: An essay on the cohesion of fluids, Philos. Trans. R. Soc., 95, 65–87, 1805.